# Forgetting-MarI: LLM Unlearning via Marginal Information Regularization

## Abstract

As AI models are trained on ever-expanding datasets, the ability to remove the influence of specific data from trained models has become essential for privacy protection and regulatory compliance. Unlearning addresses this challenge by selectively removing parametric knowledge from the trained models without retraining from scratch, which is critical for resource-intensive models such as Large Language Models (LLMs). Existing unlearning methods often degrade model performance by removing more information than necessary when attempting to "forget" specific data. We introduce *Forgetting-MarI*, an LLM unlearning framework that provably removes only the additional (marginal) information contributed by the data to be unlearned, while preserving the information supported by the data to be retained. By penalizing marginal information, our method yields an explicit upper bound on the unlearn dataset's residual influence in the trained models, providing provable undetectability. Extensive experiments confirm that our approach outperforms current state-of-the-art unlearning methods, delivering reliable forgetting and better preserved general model performance across diverse benchmarks. This advancement represents an important step toward making AI systems more controllable and compliant with privacy and copyright regulations without compromising their effectiveness.[*]

## 1 Introduction

As machine learning models, particularly Large Language Models (LLMs), get trained on bigger datasets containing potentially sensitive or regulated information, and as LLMs are increasingly deployed in high-stakes domains, the need to selectively remove specific data influences from these models has become critical. This requirement is driven not only by privacy regulations such as the European Union's General Data Protection Regulation (GDPR) and its "right to be forgotten," but also by practical concerns including the removal of copyrighted content, personally identifiable information, or data determined to be harmful or biased [32, 15, 11, 5, 3]. Unlearning, or removing the *influence* of specific data post hoc, is an attractive tool for achieving this information removal, especially with the high costs of retraining a model from scratch.

Existing unlearning methods often *over*-unlearn, removing all information linked to the data to unlearn/forget, including knowledge also legitimately supported by the data meant to be preserved. This indiscriminate approach leads to degraded model performance on tasks unrelated to the distinctive information to be forgotten.

To illustrate this distinction, consider a copyright unlearning scenario where we have an LLM pretrained on an article from *The Washington Post* and on one from *The New York Times*, but only the former is legally authorized for use. Both outlets report on an identical event, yet their articles differ in narrative style, phrasing, and editorial perspective. There are two distinct unlearning objectives with this setup:

- **Marginal Information Unlearning:** Remove only the stylistic elements, phrasing and content unique to the *Times* article, while retaining shared factual content that also appears in the authorized *Washington Post* article.

---

[*]The implementation will be released on GitHub upon acceptance of this manuscript.

- **Full Information Unlearning:** Erase all content associated with the *Times* article, including factual information that is independently supported by the retained *Washington Post* article.

We argue that the former objective is the natural objective when people talk about unlearning [and LLM unlearning naturally targets marginal unlearning.] Indeed, the goal of unlearning is not to eradicate knowledge contained in the unlearn data, but rather to surgically remove only its *marginal effect*, the information not already supported by the data we are authorized to use. In this copyright scenario, the marginal effect unlearning satisfies legal requirements with minimal utility loss, whereas the full removal would unnecessarily discard information that is lawfully present in the model.

This distinction motivates our proposed method, *Forgetting-MarI*, a direct *marginal information*[†] removal of the unlearned data. [More specifically, *marginal information unlearning* optimizes an objective that *directly* measures and suppresses only the *additional* information contributed by the unlearn set beyond what is already supported by the retain set, where the utility term only aims to stabilize retain performance or help the model learn new datasets if needed. There is no intrinsic conflict between the unlearn and utility objectives. In contrast, existing LLM unlearning methods are *full-information* in principle: their ascent or preference loss term targets the *entire* signal of $\mathcal{D}_u$ (e.g., maximizing CE on $\mathcal{D}_u$) and they attempt to *indirectly* spare shared/legitimate knowledge by counterbalancing this with a retain loss (CE or KL), preference shaping, parameter subtraction, or orthogonality (see details Appendix A.1). In other words, there is an intrinsic conflict between the utility and unlearning objective, which is necessary for the counterbalance to work, but often leads to unstable unlearning and requires extreme effort in parameter-tuning.]

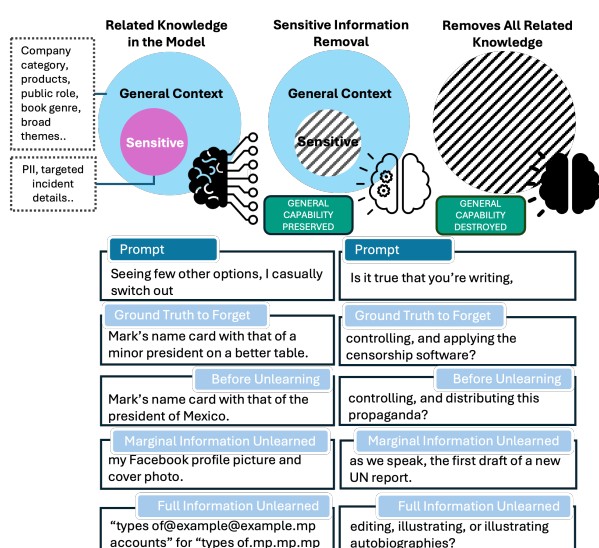

Figure 1: Comparison of sentence completions generated by Llama-3.2-1B models before and after different unlearning methods.

Figure 1 further demonstrates the difference between full-information unlearning and marginal-information unlearning (detailed experimental setup in Section 4). We created three models trained on ground truth prompts: one before unlearning, one after marginal information unlearning, and one after full information unlearning. Models are given the first half of a sentence (prompt) and are asked to complete it. Before unlearning, the model completes the sentences in a way that is similar to the ground truth. With marginal unlearning, the model produces different but coherent completions. With full unlearning, the model struggles to coherently complete the sentences.

## 1.1 Open Challenges in LLM Unlearning

Effective LLM unlearning must balance three objectives [25]. First, *unlearn efficacy* measures how well a model suppresses the influence of the data we want to unlearn, called the unlearn set $\mathcal{D}_u$. Second, *utility preservation* ensures the model's ability to retain performance on general tasks and the data we are still authorized to use, called the retain set, $\mathcal{D}_r$ is not lost. Finally, *computational cost* encompasses the time, memory, and carbon used during unlearning. All unlearning techniques aim to optimize these three objectives, which inherently come with tradeoffs; what differs is *where* and *how* the model parameters are updated, directly affecting their ability to balance the three. A breakdown of existing techniques and their strengths and weaknesses is shown in Table 1, with their technical details and commonality in indirect marginal unlearning in Appendix A.1.

Despite rapid progress, LLM unlearning is still an emerging discipline with several open challenges, summarized in Table 2.

---

[†]The term Marginal Information is formalized in Definition 1.1.

Table 1: Comparison of LLM Unlearning Approaches

| Approach | Key Methods | Strengths | Limitations |
|---|---|---|---|
| **Full tuning** | Loss-Rev.[1], GradDiff[2], KL-Dist.[3], DPO[4]/NPO[5] | Fine-grained control; preserves alignment | Risk of over-forgetting; utility drop |
| **Weight edit** | ROME[7], MEMIT[8], AlphaEdit[9], DEPN[10] | Fast; memory-light; layer-local patches | Less effective on distributed/stylistic info |
| **Counterfactuals** | IDK[11], EntityAnon.[12], ULMR[13], SKU[14] | No weight edits; easy deployment | Prompt-dependent; leakage if coverage is thin |
| **Adaptation** | Task-vec[15], O3[16] | Modular patches; base model unchanged | Adapters grow linearly; gate failures leak |

[†]Numbers map to BibTeX entries: [1][42], [2][25], [3][17], [4][33], [5][43], [7][30], [8][31], [9][10], [10][39], [11][20], [12][9], [13][36], [14][26], [15][19], [16][12].

*Robust unlearning & Utility Preservation*: Existing LLM unlearning techniques via full-parameter fine-tuning typically treat the unlearn set $\mathcal{D}_u$ as fully toxic, forcing the model to forget every sequence in $\mathcal{D}_u$ regardless of their overlap with the retain set $\mathcal{D}_r$. Examples include loss-reversal [24], gradient-difference [42], KL-ascent [17], and preference-based DPO/NPO [33, 43]. Even local editors that aim to make precise edits (ROME, MEMIT) share

| Method | Utility Preservation | Scalable Continual | Formal Guarantees |
|---|---|---|---|
| Full Parameter | ☆ | ✓ | ✗ |
| Weight Editing | ☆ | ✓ | ✗ |
| Counterfactual | ✗ | ✓ | ✗ |
| Adaptation | ☆ | ☆ | ✗ |
| **Ours** | ✓ | ✓ | ✓ |

Table 2: Comparison of families of unlearning methods based on literature evidence. Our proposed marginal effect unlearning addresses key limitations of existing approaches. (✓=yes, ✗=no, ☆=partial)

this limitation [30, 31], erasing shared facts and stylistic cues, and raising perplexity on $\mathcal{D}_r$ and held-out tasks. Benchmarks (RWKU, MUSE, Eight-Method) consistently report sizable utility drops after unlearning [21, 44, 27].

*Stable Continual Unlearning*: As the legal landscape around data usage changes, a deployed LLM may receive hundreds or thousands of unlearn requests. Production-ready unlearning, therefore, needs to be able to repeatedly unlearn, retain utility, and keep computation and memory within a practical range. Exact methods like full retraining or shared SISA guarantee unlearning but their cost scales with both model size and request count [2, 14, 1]. Lighter updates like influence functions [16] or repeated ROME/MEMIT edits [30, 31] are cheap per removal yet accumulate inference costs and utility drift. Task-vector subtraction or adapter stacks save compute during unlearning but require storing external model adapters, also creating downstream inference costs [19, 12, 38]. Thus, continually unlearning without runaway resources or utility loss remains unsolved.

*Formal Guarantees at LLM-Scale*: Certified unlearning is well established for linear/kernel models [16], high-dimensional classifiers [6], and general mathematical formulations of machine unlearning [41]. However, no existing method provides guarantees that scale to autoregressive transformers with billions of parameters (7B–70B+), such as GPT or Llama. As a result, practitioners lack *reliable guarantees* of the extent to which the unlearn set remains uninferable or undetectable after common downstream operations such as compression, distillation, or adversarial probing [25].

## 1.2 OUR CONTRIBUTIONS

To address these challenges, we introduce *Forgetting-MarI*, a novel information-theoretic LLM unlearning framework. First, we provide a heuristic definition of marginal information (formal quantification appears in Section 2.1):

**Definition 1.1** (Marginal Information (MarI)). *Marginal information is the marginal effect on model inference when adding the unlearn set to the retain set.*

The core idea of Forgetting-MarI is to penalize the model in proportion to the marginal information, and thus eliminate only the *unique* contribution of the unlearn dataset on the model's parameters and its inference abilities. This avoids erasure of shared information between the retain and unlearn sets. A key piece of our technique, therefore, is an accurate quantification of marginal information, which we detail in Section 2.1. Forgetting-MarI can be summarized by the following learning objective:

$$\min_{\text{model parameter: } \theta} \ell_{\text{utility}}(\text{model}(\theta), \mathcal{D}_r) + \ell_{\text{MarI}}(\text{model}(\theta), \mathcal{D}_r, \mathcal{D}_u),$$

with $\ell_{\text{utility}}$ being a loss that aims to maintain the utility of the model and $\ell_{\text{MarI}}$ being the marginal information loss derived from an accurate marginal information quantification.

The key contributions of our proposed method include:

- **(A1)** *Utility preservation*: Targeting marginal information means that only the marginal effect of the unlearn set is removed, preserving information shared with the retain set.
- **(A2)** *Scalable and continual*: Using an additive mutual-information regularizer integrates with standard gradient-based fine-tuning and naturally supports continual unlearning.
- **(A3)** *Theoretical unlearning guarantee*: Bounding marginal information yields an explicit upper bound on residual mutual information, providing provable undetectability of the unlearn set.
- **(A4)** *Exemplary experimental performance:* Experiments show that our proposed method outperforms state-of-the-art unlearning methods in unlearning tasks using [real-world text data on mid-scale LLMs].

## 2 UNLEARNING: MARGINAL INFORMATION

Forgetting-MarI relies on a novel quantification of *marginal information* that (i) vanishes when the unlearn set $\mathcal{D}_u$ adds *no new* information beyond the retain set $\mathcal{D}_r$, and (ii) increases as $\mathcal{D}_u$ contributes information absent from $\mathcal{D}_r$, recovering the full information in $\mathcal{D}_u$ as $\mathcal{D}_r$ vanishes. We propose a mutual information (MI)–based quantification that satisfies these properties.

### 2.1 QUANTIFYING AND UNLEARNING MARGINAL EFFECTS

Fix a language model $p_\theta$ (with parameter $\theta$) over a finite vocabulary $V$ and a length $T \geq 1$. For $y \in V^T$, let $p_\theta(\cdot \mid y_{<t})$ be the next-token distribution. For a subset $s \subseteq V^T$, let $\mu_s$ be the uniform law on $s$ and define its averaged next-token marginals $(p_\theta)^s_t(v) := \mathbb{E}_{Y \sim \mu_s}\big[p_\theta(v \mid Y_{<t})\big]$ for $t \in [T]$, $v \in V$. Write $p^r := \{(p_\theta)^r_t\}_{t \in [T]}$, $p^u := \{(p_\theta)^u_t\}_{t \in [T]}$. For $d := r \cup u$, $p^d_t = \alpha\, p^r_t + (1-\alpha)\, p^u_t$, $\alpha := \frac{|r|}{|r|+|u|} \in (0,1)$. Let $T^* \sim \text{Uniform}([T])$ and $Z \sim \text{Bernoulli}(\frac{1}{2})$ be independent. Conditioned on $(T^* = t, Z)$, draw $X \sim p^d_t$ if $Z = 0$ and $X \sim p^r_t$ if $Z = 1$, and set $X_{\text{MarI}} := (T^*, X)$. Then the *mutual information* between $X_{\text{MarI}}$ and $Z$ is defined as

$$I(X_{\text{MarI}}; Z) := \frac{1}{T} \sum_{t=1}^{T} \text{JSD}\big(p^d_t, p^r_t\big). \tag{1}$$

Here, we denote the Jensen-Shannon divergence as $\text{JSD}(p, q) := \frac{1}{2} D_{\text{KL}}(p\|m) + \frac{1}{2} D_{\text{KL}}(q\|m)$ with $m := \frac{p+q}{2}$ and $D_{\text{KL}}(p\|q) := \sum_v p(v) \log \frac{p(v)}{q(v)}$. By construction, the information or distribution represented by $d$ can be decomposed into the contribution of $r \cap d = r$ and the marginal contribution of $d \setminus r = u$. The distribution contributed by $r$ through the model $p_\theta$ is $p^r$. The distributional contribution from the addition of $u$ through $p_\theta$ is the distributional difference between $p^r$ and $p^d$.

By construction, the quantification of the marginal effect is small if $p^r$ is close to $p^d$, because such proximity suggests that the information content in $u$ has already been largely represented by $r$. Conversely, the quantification will be large if $p^r$ differs significantly from $p^d$, indicating that $u$ contributes substantial new information w.r.t. $p_\theta$ and induces a model output distribution shift. Therefore, defining this marginal effect quantification boils down to differentiating $p^r$ from $p^d$ for any $r \subset \mathcal{D}_r$ and $d = r \cup u$ with arbitrary $u \subset \mathcal{D}_u$.

A natural way to *quantify* this difference is via a binary detection problem. Consider a binary detection problem using the construction above:

$$X_t := X\big|_{T^*=t} \sim \begin{cases} p^d_t, & Z = 0, \\ p^r_t, & Z = 1, \end{cases} \qquad \mathbb{P}[Z = 0] = \mathbb{P}[Z = 1] = \tfrac{1}{2}. \tag{2}$$

If $p^r = p^d$, even an optimal classifier does no better than a coin flip. If there is distributional shift, it can detect the difference. A sharp information–theoretic upper bound on the Bayes accuracy, denoted by $P_{\text{acc}}$ and defined below in Proposition 2.1, is the following:

**Proposition 2.1** (Detection accuracy upper bounded by mutual information). *For $(X_{\mathrm{MarI}}, Z)$ with prior $\pi = \mathbb{P}[Z = 1]$,*

$$P_{\mathrm{acc}} \;=\; \mathbb{E}\left[\max\{P(Z = 0\,|\,X_{\mathrm{MarI}}),\, P(Z = 1\,|\,X_{\mathrm{MarI}})\}\right] \;\leq\; 1 - H_2^{-1}\big(H_2(\pi) - I(X_{\mathrm{MarI}}; Z)\big),$$

*where $H_2(\cdot)$ is the binary entropy and $H_2^{-1}$ denotes the inverse of $H_2$ restricted to $[0, \frac{1}{2}]$.*

Proof in Appendix B.1. Here $P(Z \,|\, X_{\mathrm{MarI}})$ denotes the Bayes-optimal posterior between retain $r$ and union $d$. Note that $I(X_{\mathrm{MarI}}; Z) \in [0, H_2(\pi)]$ satisfies: (i) $I(X_{\mathrm{MarI}}; Z) = 0$ when $p^d = p^r$; (ii) $I(X_{\mathrm{MarI}}; Z)$ grows with their divergence, approaching $H_2(\pi)$ as $\mathcal{D}_r$ vanishes. Since $p^d \neq p^r$ occurs precisely when $u$ induces model confidence shifts not explained by $r$, Proposition 2.1 gives $I(X_{\mathrm{MarI}}; Z)$ an intuitive meaning as the detectability of the marginal effect (Definition 1.1).

**Definition 2.1** (MI-based marginal information loss). *With $(X_{\mathrm{MarI}}, Z)$ as in equation 2, define*

$$\ell_{\mathrm{MarI}}(\theta, r, u) \;:=\; I(X_{\mathrm{MarI}}; Z).$$

Thus, Forgetting-MarI solves

$$\min_{\theta} \;\; \ell_{KL}(\theta, r) \;+\; \ell_{\mathrm{MarI}}(\theta, r, u), \tag{3}$$

where $\ell_{KL}(\theta, r) := D_{\mathrm{KL}}\big(p^r(\theta) \,\|\, p^r(\theta_0)\big)$ is the KL divergence between the updated model (parameter $\theta$) and the frozen original model (parameter $\theta_0$) on $r$, and $\ell_{\mathrm{MarI}}$ is as above, motivated by Prop. 2.1. Algorithm 1 describes an efficient LLM implementation.

**Remark 2.1** (Alternative quantification). *Marginal information measures the shift from $p_\theta(r)$ to $p_\theta(d)$. Alternatively, one may use $\ell'_{\mathrm{MarI}}(\theta, r, u) := D_{\mathrm{KL}}\big(p^d \,\|\, p^r\big)$ or $D_{\mathrm{KL}}\big(p^d \,\|\, p^r(\theta_0)\big)$. But mutual information has the advantage of (1) stability (boundedness), (2) interpretability (Proposition 2.1), and (3) continuous unlearning (evolving reference $m = \frac{p+q}{2}$). See Appendix B.2 for details.*

## 2.2 MARGINAL INFORMATION & PERPLEXITY-BASED DETECTORS

We provide theoretical guarantees for the unlearning performance of Forgetting-MarI against white-box copyright detectors that rely on model confidence (perplexity / cross-entropy). Let

$$S_\theta(x, y) \;=\; \frac{1}{T} \sum_{t=1}^{T} \big(-\log p_\theta(x_t \,|\, y_{<t})\big)$$

be the standard cross-entropy (per-token negative log-likelihood). State-of-the-art detectors [3, 45, 29] flag the membership of $x$ in training by testing whether $S_\theta(x, x)$ is suspiciously low. We adopt the notation from Section 2.1: sequences $r, u \in V^T$, next-token marginals $p_t^r, p_t^u$, their mixture $p_t^d = \alpha\, p_t^r + (1 - \alpha)\, p_t^u$ with $\alpha = \frac{|r|}{|r|+|u|}$, and the mutual information $I(X_{\mathrm{MarI}}; Z)$.

The next result shows that, given a set of sequences to forget, denoted by $u$, Forgetting-MarI guarantees that there is a set of sequences in the retain set, denoted by $r$, such that the score $S_\theta(u, u)$ is close to $S_\theta(u, r)$. In other words, a high model confidence implied by $S_\theta(u, u)$ is possibly due to the existence of $r$ because one would get the same score for $u$ if feeding the model $r$ instead of $u$.

**Theorem 2.1** (MarI controls the self-perplexity gap). *Fix $u = (u_1, \ldots, u_T) \in V^T$ and assume the pathwise non-vanishing condition $\min\{p_t^u(u_t),\, p_t^r(u_t)\} \geq \gamma \in (0, 1]$ for all $t$. Then*

$$\left| S_\theta(u, u) - S_\theta(u, r) \right| \;\leq\; \frac{2\sqrt{2}}{\gamma(1 - \alpha)} \; \sqrt{I(X_{\mathrm{MarI}}; Z)}.$$

See Appendix B.3 for the proof. In particular, when $I(X_{\mathrm{MarI}}; Z)$ goes to zero, the score gap above vanishes, and the perplexity/log-likelihood detectors lose discriminative power after unlearning.

Finally, we show that MarI directly controls the score gap even for a neighborhood of $u$ rather than only $u$ itself. [The formal result and its proof] can be found in Appendix B.4.

---

**Algorithm 1:** Forgetting-MarI

**Require:** Retain dataset $\mathcal{D}_r = \{x_i^r\}_{i=1}^{|\mathcal{D}_r|}$, unlearn dataset $\mathcal{D}_u = \{x_j^u\}_{j=1}^{|\mathcal{D}_u|}$; initial model $f_{\theta_0}$;
      learning rate $\eta$; batch sizes $B_r, B_u$; epochs $E$; trade-off $\gamma \in (0, 1)$;
      hetero $\in \{\texttt{False}, \texttt{True}\}$.
**Ensure:** Unlearned parameters $\theta_E$.
  1: Initialize $\theta \leftarrow \theta_0$; build dataloaders $\mathcal{L}_r, \mathcal{L}_u$.
  2: **for** $e = 1$ **to** $E$ **do**
  3:    Shuffle $\mathcal{L}_r, \mathcal{L}_u$; $S \leftarrow \min\{|\mathcal{L}_r|, |\mathcal{L}_u|\}$.
  4:    **for** $s = 1$ **to** $S$ **do**
  5:       **Minibatches:** $r \leftarrow \mathcal{L}_r(s), \quad u \leftarrow \mathcal{L}_u(s)$.
  6:       **Model logits & probabilities:**
  7:         $L^r \leftarrow f_\theta(r) \in \mathbb{R}^{B_r \times T \times |V|}, \; p_{b,t}^r \leftarrow \mathrm{softmax}(L^r[b, t, :]) \in [0, 1]^{B_r \times T \times |V|}$;
  8:         $L_0^r \leftarrow f_{\theta_0}(r) \in \mathbb{R}^{B_r \times T \times |V|}, \; p_{b,t}^r(\theta_0) \leftarrow \mathrm{softmax}(L_0^r[b, t, :]) \in [0, 1]^{B_r \times T \times |V|}$;
  9:         $L^u \leftarrow f_\theta(u) \in \mathbb{R}^{B_u \times T \times |V|}, \; p_{b,t}^u \leftarrow \mathrm{softmax}(L^u[b, t, :]) \in [0, 1]^{B_u \times T \times |V|}$;
10:         $p^d \leftarrow \mathrm{concat}_{\mathrm{batch}}([p^r, p^u]) \in [0, 1]^{(B_r + B_u) \times T \times |V|}$.
11:       **Utility Loss:**
12:         $\bar{p}^r \leftarrow \frac{1}{B_r * T} \sum_{b,t} p_{b,t}^r, \; \bar{p}_0^r \leftarrow \frac{1}{B_r * T} \sum_{b,t} p_{b,t}^r(\theta_0), \quad \ell_{\mathrm{KL}} \leftarrow \sum_{v \in V} \bar{p}^r(v) \log(\frac{\bar{p}^r(v)}{\bar{p}_0^r(v)})$.
13:       **MarI Loss:**
14:         $P(v, 0) \leftarrow \frac{1}{2(B_r + B_u) * T} \sum_{b,t} p_{b,t}^d, \; P(v, 1) \leftarrow \frac{1}{2 B_r * T} \sum_{b,t} p_{b,t}^r, \; P(v) = \sum_{z \in \{0,1\}} P(v, z)$,
15:         $\ell_{\mathrm{MarI}} \leftarrow \sum_{z \in \{0,1\}} \sum_{v \in V} P(v, z) \log \frac{P(v,z)}{P(v) P(z)}$.
16:       **Total loss:** $\mathcal{L} = (1 - \gamma) \ell_{\mathrm{KL}} + \gamma \ell_{\mathrm{MarI}}^{\mathrm{flat}}$.
17:       **Update:** $\theta \leftarrow \theta - \eta \nabla_\theta \mathcal{L}$.
18:    **end for**
19: **end for**
20: **return** $\theta_E \leftarrow \theta$

---

Figure 2: Pseudo-code for Forgetting-MarI.

## 3 ALGORITHM DESIGN

*Token-wise* MarI (Equation 1) provides strong guarantees when sentences in $r$ and $u$ are *homogeneous* in length and token-wise context. In practice, however, token-wise MarI Loss can be noisy under heterogeneous batches. To address this, we also provide a *pooled* ("flattened") estimator that first averages across token positions (and batch) to form $\bar{p}^s = \frac{1}{T} \sum_t p_t^s$, $s \in \{r, u, d\}$, then computes the pooled MarI Loss $I(\bar{X}_{\mathrm{MarI}}; Z) = \mathrm{JSD}(\bar{p}^d, \bar{p}^r)$. Such a pooled version aims to stabilize the marginal information quantification by filtering the position-heterogeneous noise and emphasizing the dominant distribution shift.

By the data-processing inequality, the pooled estimator $I(\bar{X}_{\mathrm{MarI}}; Z)$ is a *variational lower bound* to the token-wise MarI. The gap between the pooled and token-wise MarI losses is controlled by the $\ell^2$-deviation of the token-sequence densities (details in Theorem C.1[Pooling error bound], Appendix C). [Furthermore, pooled MarI offers a specific word-wise guarantee:

**Theorem 3.1** (Word-level provable unlearning via pooled MarI). *Fix a token* $w \in V$ *and assume* $\bar{p}^u(w) \wedge \bar{p}^r(w) =: \bar{\gamma}_w \in (0, 1]$. *Then*

$$\left| \log \bar{p}^u(w) - \log \bar{p}^r(w) \right| \leq \frac{2\sqrt{2}}{\bar{\gamma}_w (1 - \alpha)} \sqrt{I(\bar{X}_{\mathrm{MarI}}; Z)}.$$

Proof in Appendix C. Although weaker than the sentence-wise guarantee (Theorem 2.1), this guarantee is sufficient for provable unlearning of specific tokens and can be useful for the provable removal of word-level (not related to sentence structure) information.]

[Figure 2 presents pseudo-code for Forgetting-MarI with pooled MarI. A full, detailed algorithm is provided as Figure 1 in Appendix C. We recommend the following protocol:

- **Estimator Selection:** Use *token-wise* MarI for homogeneous data (aligned contexts) to leverage precise signals. Use *pooled* MarI for large corpora with random batching to ensure stability.
- **Hyperparameter Tuning:** Fix a tolerable perplexity gap for the application, derive the required MarI threshold, and tune $\gamma$ until the regularization magnitude falls below this threshold. This invokes the guarantees of Theorem 2.1 or 3.1.

As both estimators achieve comparable empirical results, Section 4 reports findings under a unified label (comparative ablation in Appendix C).]

## 4 Experiments

Our experiments are designed to address three questions:

1 *Utility-Forgetting Trade-off:* Does Forgetting-MarI balances performance preservation on $\mathcal{D}_r$ while removing $\mathcal{D}_u$? By achieving similar unlearn performance $\mathcal{D}_u$ and utility preservation on $\mathcal{D}_r$ to the unlearn baseline (i.e., the retrain on retain model)?

2 *Continual Unlearning:* Is the method robust to sequential deletion requests without incurring catastrophic forgetting?

3 *Detectability & Capacity:* Consistent with our theory, does the model defeat perplexity-based detectors while retaining general capabilities?

We compare against state-of-the-art full-parameter unlearning baselines. Table 3 below summarizes these baselines alongside other unlearning approaches.

| Method | Unlearn objective | Retain objective | Unlearn nature |
|---|---|---|---|
| GA [42] | ascent on unlearn set | — | full information |
| GD [24] | ascent on unlearn set | descent on retain set | full information |
| KL-GA [17] | ascent on unlearn set | minimize $\mathrm{KL}\big(p_\theta^r \,\|\, p_{\theta_0}^r\big)$ | full information |
| DPO [33] | preference loss | minimize CE or KL | full information |
| **Forgetting–MarI** | minimize $I(X_{\mathrm{MarI}}; Z)$ | minimize $\mathrm{KL}\big(p_\theta^r \,\|\, p_{\theta_0}^r\big)$ | marginal information |

Table 3: Comparison of LLM unlearning objectives. CE = cross-entropy; KL = Kullback–Leibler divergence. "Direct/Indirect" indicates whether the method explicitly penalizes marginal information (ours) or approximates it by balancing forget/retain signals.

### 4.1 Models, Datasets, and Splits

**Protocol.** Following Maini et al. [28], Eldan & Russinovich [9], we adopt a three-step evaluation: (i) fine-tune on $\mathcal{D}_u \cup \mathcal{D}_r$ to obtain a *full finetune baseline*; (ii) apply each unlearning method to remove the influence of $\mathcal{D}_u$ to obtain unlearn results, denoted by the name of corresponding unlearn methods; (iii) restart the fine-tune but only on $\mathcal{D}_r$ (never exposed to $\mathcal{D}_u$) to obtain an *unlearn baseline*. An ideal unlearning method should preserve retain/validation accuracy level similar to the unlearn baseline while lowering the unlearn accuracy to a similar level to the unlearn baseline. We report next-token accuracy on $\mathcal{D}_r$, $\mathcal{D}_u$, and a held-out validation set, and assess general capability with the Eleuther's LM Evaluation Harness [13].

**Datasets.** We evaluate Forgetting–MarI on two mid-scale LMs, GPT-2 Large (774M) and Llama-3.2-1B, and two text domains with distinct genres and pretraining prevalence: (i) *Harry Potter and the Prisoner of Azkaban* [22], likely present in pretraining; and (ii) *Careless People: A Cautionary Tale of Power, Greed, and Lost Idealism* [40], published after Llama's release and thus unlikely to appear in its pretraining.

**Splits.** For *Harry Potter* (HP), we designate 10% of sentences as $\mathcal{D}_u$ and the remaining 90% as $\mathcal{D}_r$ (cf. 9); validation uses excerpts from another book in the series (*Harry Potter and the Sorcerer's Stone* [34]). For *Careless People* (CP), we consider: (i) a *correlated* split with contiguous 50/50 spans for $\mathcal{D}_u/\mathcal{D}_r$; and (ii) an *uncorrelated* split where $\mathcal{D}_u$ comprises 2025 Reddit stories (post-release) and $\mathcal{D}_r$ is 50% of the book. Reddit stories are used as validation in the first setting and the remaining 50% of the book is used as validation in the second setting.

[**Parameter tuning & ablation.** For each method, we stop training once validation accuracy drops by more than 3% from its initial value (to prevent general-utility degradation). Hyperparameter sweeps, ablations, and full training trajectories are provided in Appendix D.4.]

[**Implementation details** for hardware specifications and runtime information can be found in Appendix D.1.]

### 4.2 Accuracy Trade-off & general capability

[Below, unlearn baseline is the model fine-tuned on retain set only, which serves as a retrain from scratch model.]

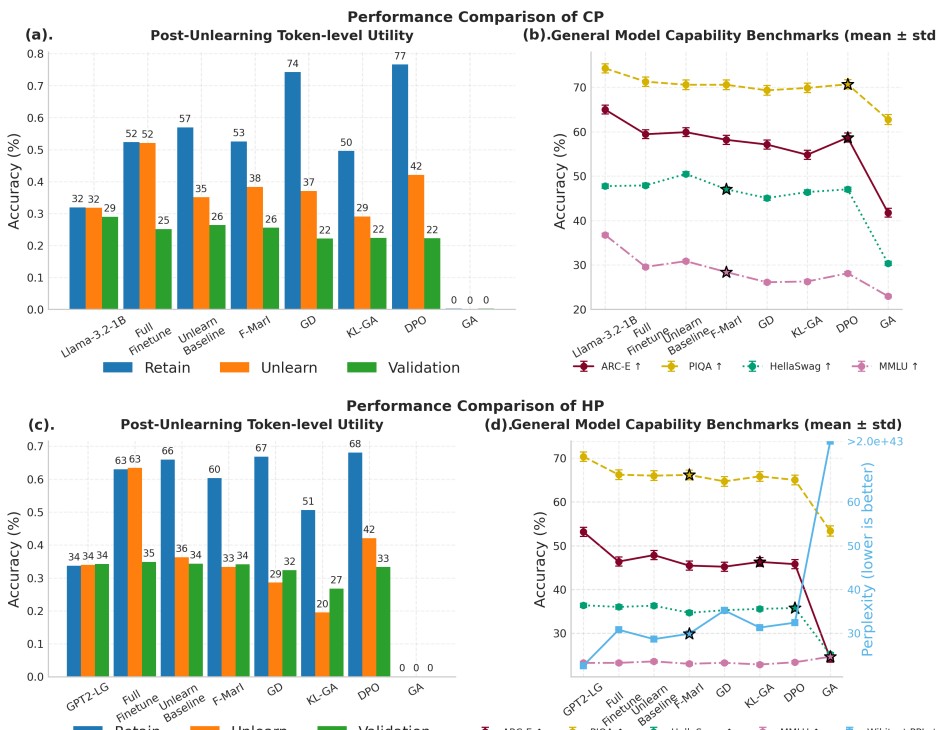

Figure 3: [Left panels summarize the next-token accuracies on retain/unlearn/validation whereas the right panels summarize the general-capability on various benchmarks. Top row shows the results from Llama-3.2-1B on *Careless People* (correlated split), where as bottom row shows the results from GPT-2 Large on *Harry Potter*. Each method is reported at its best $\lambda$ and training epoch. On the left panels, an ideal method should match the unlearn baseline on retain/unlearn/validation accuracy on the left panels. On the right, better methods should achieve higher accuracy on ARC-E, HellaSwag, PIQA, MMLU and lower on WikiText perplexity test. Star indicates the best performer on that test.]

[ **Token-level utility (left panels).** Across HP and CP, an ideal unlearning method should achieve similar retain/unlearn/validation accuracy as the unlearn baseline. Forgetting–MarI closely matches the unlearn baseline by achieving similar retain/unlearn/validation accuracy in both HP and CP experiments. In comparison, we observe unstable behaviors from methods with full-information unlearning nature: *(i) Careless People*: GD and DPO both overtrain on $\mathcal{D}_r$, while KL-GA struggles to remove the information in $\mathcal{D}_u$ while maintaining performance on $\mathcal{D}_r$. Moreover, all other methods show a gradual degradation on the validation accuracy. (The information of uncorrelated retain/unlearn split experiment is in Appendix D.) *(ii) HP*: GD and DPO are less overtrain on $\mathcal{D}_r$ but either over- or under-unlearn compared to the unlearn baseline. KL-GA still struggles to remove the information in $\mathcal{D}_u$ while maintaining performance on $\mathcal{D}_r$.]

[**General capability (right panels).** We compare the unlearned models against the finetuned baseline, the unlearn baseline, and the original GPT-2 (before finetuning) on ARC-E, PIQA, HellaSwag, WikiText, and MMLU [13].

- *Llama on Careless People.* Forgetting–MarI attains the best results on HellaSwag and MMLU and ranks second on PIQA and ARC-E (DPO is first there but fails to unlearn effectively). It is also the only method that largely matches the full finetune baseline, which implies no model capacity degradation, because unlearning starts from full finetune baseline. Furthermore, notice that full finetuned baseline has lower general capacity than the unlearn baseline. That suggests a better finetuning could help Forgetting-MarI match the unlearn baseline across all benchmarks.
- *GPT-2 on Harry Potter.* Forgetting–MarI is best on PIQA and WikiText, second on ARC-E, and slightly behind KL-GA/GD on HellaSwag. Furthermore, we note that high scores on general benchmarks do not guarantee meaningful utility: For instance, GA can outperform all other methods on MMLU despite lacking any utility preservation (and exhibiting very high WikiText perplexity which is consistent with theory). Thus, evaluating perplexity or token-level confidence is essential to avoid confounding factors.

| Method | Unlearning Step | Retain (%) ↑ | Validate (%) ↑ | Hermione (%) ↓ | Snape (%) ↓ | Ron (%) ↓ | ARC-E (%) ↑ | WikiText (PPL) ↓ |
|--------|-----------------|--------------|----------------|----------------|-------------|-----------|-------------|------------------|
| KL-GA | Step 1: Forget Hermione | 51.0 | 33.5 | 4.7 | 58.0 | 59.6 | 46.1 | 33.7 |
| KL-GA | Step 2: Forget Snape | 52.2 | 34.2 | 9.4 | 9.2 | 61.8 | 46.2 | 50.0 |
| KL-GA | Step 3: Forget Ron | 49.3 | 28.9 | 9.6 | 14.7 | 7.1 | 44.6 | 3.4e+10 |
| | | | | | | | | |
| DPO | Step 1: Forget Hermione | 67.3 | 33.3 | 44.6 | 69.0 | 69.5 | 46.4 | 33.1 |
| DPO | Step 2: Forget Snape | 69.9 | 33.5 | 45.0 | 46.4 | 71.9 | 45.5 | 35.6 |
| DPO | Step 3: Forget Ron | 79.9 | 33.4 | 42.0 | 46.0 | 50.4 | 46.2 | 40.3 |
| | | | | | | | | |
| GD | Step 1: Forget Hermione | 66.3 | 33.3 | 11.7 | 67.2 | 67.7 | 44.1 | 46.3 |
| GD | Step 2: Forget Snape | 68.8 | 33.2 | 12.0 | 15.7 | 70.1 | 43.1 | 62.4 |
| GD | Step 3: Forget Ron | 78.2 | 31.7 | 10.7 | 15.8 | 16.0 | 42.8 | 61550 |
| | | | | | | | | |
| F-MarI | Step 1: Forget Hermione | 60.0 | 36.3 | 16.2 | 60.9 | 62.9 | 46.1 | 29.5 |
| F-MarI | Step 2: Forget Snape | 60.3 | 36.2 | 15.5 | 14.0 | 63.1 | 45.2 | 29.9 |
| F-MarI | Step 3: Forget Ron | 60.4 | 34.9 | 13.4 | 15.0 | 15.2 | 44.0 | 38.5 |

(a) GPT2-LG on *Harry Potter* (HP).

| Method | Unlearning Step | Retain (%) ↑ | Validate (%) ↑ | First 15 (%) ↓ | Next 15 (%) ↓ | Next 15 (%) ↓ | ARC-E (%) ↑ | PIQA (%) ↑ |
|--------|-----------------|--------------|----------------|----------------|---------------|---------------|-------------|------------|
| KL-GA | Step 1: Forget First 15% | 43.1 | 16.3 | 7.5 | 32.2 | 34.0 | 53.7 | 68.9 |
| KL-GA | Step 2: Forget Second 15% | 31.1 | 5.3 | 3.0 | 1.8 | 12.0 | 53.2 | 67.3 |
| KL-GA | Step 3: Forget Third 15% | 21.9 | 2.4 | 1.5 | 1.1 | 0.9 | 52.1 | 66.4 |
| | | | | | | | | |
| DPO | Step 1: Forget First 15% | 75.0 | 22.9 | 42.5 | 46.1 | 46.5 | 57.2 | 70.7 |
| DPO | Step 2: Forget Second 15% | 82.6 | 20.8 | 40.4 | 38.3 | 42.8 | 54.9 | 70.3 |
| DPO | Step 3: Forget Third 15% | 85.9 | 20.1 | 39.1 | 37.4 | 36.2 | 52.0 | 68.6 |
| | | | | | | | | |
| GD | Step 1: Forget First 15% | 76.5 | 19.3 | 21.3 | 41.4 | 41.4 | 53.8 | 69.5 |
| GD | Step 2: Forget Second 15% | 74.7 | 12.7 | 15.4 | 7.0 | 30.2 | 48.9 | 68.3 |
| GD | Step 3: Forget Third 15% | 76.5 | 8.6 | 11.5 | 5.8 | 3.8 | 41.5 | 64.2 |
| | | | | | | | | |
| F-MarI | Step 1: Forget First 15% | 51.9 | 25.3 | 35.0 | 49.7 | 50.0 | 58.6 | 70.7 |
| F-MarI | Step 2: Forget Second 15% | 51.6 | 25.1 | 34.7 | 30.4 | 47.7 | 57.9 | 70.6 |
| F-MarI | Step 3: Forget Third 15% | 51.2 | 25.0 | 34.3 | 30.2 | 27.8 | 55.9 | 69.3 |

(b) Llama-3.2-1B on *Careless People* (CP).

Figure 4: [Continual unlearning across models and datasets. Rows show methods and sequential steps; columns report retain/validation accuracies, performance on chunk-specific content, and general capability. Darker color indicates higher accuracy. For general capabilities (ARC-E, PIQA and WikiText): higher accuracies/lower perplexities and stable performance across steps indicates successful knowledge preservation. For unlearned content: ideal pattern shows accuracy drop after removal, with persistent forgetting in subsequent steps.]

In summary, Forgetting–MarI delivers targeted forgetting while largely preserving general capability, outperforming full-information baselines on the retain/unlearn/validation trade-off. See complete benchmark tables for experiment results on ARC-E, PIQA, MMLU, HellaSwag, and WikiText in Appendix D.5 (Tables 5 and 6).]

### 4.3 CONTINUAL UNLEARNING WITH SEQUENTIAL DELETION REQUESTS

**Setup.** [For continual unlearning, we adopt a multi-stage protocol: we partition the total forget set into three disjoint subsets $\mathcal{D}_u = \mathcal{D}_{u,1} \cup \mathcal{D}_{u,2} \cup \mathcal{D}_{u,3}$. Starting from a baseline model fine-tuned on the full union $(\mathcal{D}_r \cup \mathcal{D}_u)$, we perform three sequential unlearning steps: at step $t$, we unlearn $\mathcal{D}_{u,t}$ from the model resulting from step $t-1$. The subsets are defined as follows: (1) Harry Potter (Character-based): To emulate user-level unlearning, we build an HP "forget-characters" benchmark: we assign sentences to characters via alias matching (e.g., "Hermione", "Hermione Granger", "Granger" → Hermione). We target three characters sequentially: $\mathcal{D}_{u,1}$ (Hermione) → $\mathcal{D}_{u,2}$ (Snape) → $\mathcal{D}_{u,3}$ (Ron). The retain set $\mathcal{D}_r$ includes all "other" sentences, all retain-character sentences (e.g., Harry, Dumbledore), and sentences of not-yet-unlearned characters. (2) Careless People (Random-split): We randomly partition the designated forget set into three equal, disjoint subsets to form $\mathcal{D}_{u,1}$, $\mathcal{D}_{u,2}$, and $\mathcal{D}_{u,3}$. The retain set $\mathcal{D}_r$ consists of the remaining text from the book.]

[**Results.** Figures 4 summarize GPT-2/HP and Llama/CP results, respectively:

- *GPT-2 on HP.* Forgetting–MarI is the only method that remains robust across retain and validation accuracy, preserves forgetting on previously unlearned sets, and sustains general capability. By contrast, KL-GA tends to *relearn* previously forgotten content and substantially degrades Wiki-Text performance; DPO fails to effectively unlearn; GD overshoots on $\mathcal{D}_r$ and also loses general capability (WikiText).
- *Llama on CP.* Forgetting–MarI again exhibits the most consistent behavior: retain/validation accuracy and previously unlearned sets remain stable, and ARC-E/PIQA show minimal drift. In comparison, KL-GA quickly over-unlearns, lowering retain/validation and prior-unlearn accuracies; DPO again fails to unlearn effectively; GD overshoots on $\mathcal{D}_r$, reduces validation accuracy, and shows deteriorating ARC-E/PIQA performance.

**Conclusion.** Forgetting–MarI delivers robust sequential unlearning: it preserves utility, maintains prior forgetting, and sustains general capability across steps, outperforming full-information baselines in both performance and stability.]

## 4.4 Detector evaluation: Empirical verification of theoretical guarantees

Theorems 2.1 implies that, after Forgetting-MarI, the mutual information between logits and the "seen/unseen" bit $Z$ is negligible; hence *any* confidence-based test (perplexity, cross-entropy, log-likelihood, etc.) should fail to separate forgotten from genuinely unseen text.

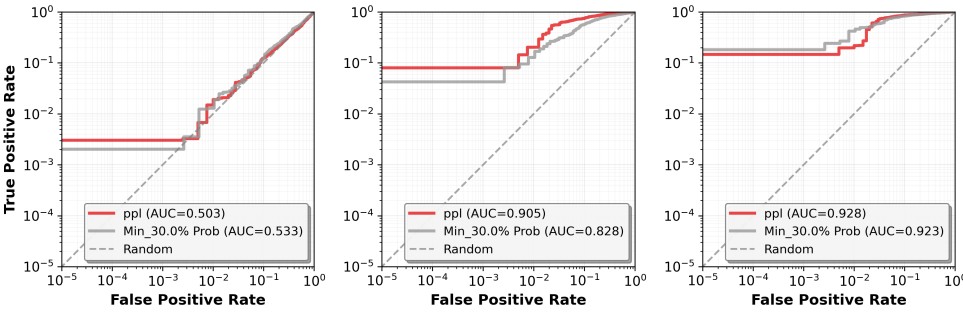

Figure 5: Detector performance for a GPT2-LG model without unlearning (left), unlearned with Forgetting-MarI (middle), and the golden-standard unlearn baseline (right). Here, ppl = perplexity.

Copyrighted-text detection methods fall into two families: (i) **white-box** detectors [3, 45, 29, 37], which use (tail or reference-model) perplexity to infer training membership; (ii) **black-box** detectors [23, 4, 8, 7, 18], which rely on string-level similarity without logits. Because Forgetting-MarI (and most threat models) allow weight access, white-box tests are strictly harder to defeat; we thus focus on them. (See Appendix D for method details and additional results.)

We ran the current SOTA white-box detector [37] on: (i) the model finetuned on $\mathcal{D}_r \cup \mathcal{D}_u$, (ii) the *gold-standard* unlearn baseline trained only on $\mathcal{D}_r$, and (iii) model (i) after Forgetting-MarI. We report ROC–AUC: low values indicate the detector believes the model was trained with $\mathcal{D}_u$, high values indicate the opposite. As shown in Fig. 5, the ROC–AUC after Forgetting-MarI closely matches the unlearn baseline, indicating effective removal of $\mathcal{D}_u$'s influence, as predicted by theory.

## 5 Conclusion

This work presents Forgetting-MarI, a novel approach to LLM unlearning that improves upon existing state-of-the-art unlearning methods while providing rigorous theoretical guarantees. Our experimental results across multiple benchmarks confirm the practical effectiveness of our proposed technique, while our theoretical analysis establishes formal bounds on the unlearning process and convergence properties.

The combination of strong empirical results and theoretical foundations represents a significant advancement in machine unlearning for LLMs. However, several important directions remain for future research. First, while our theoretical guarantees provide valuable insights into the method's behavior, there is still a gap between theoretical bounds and the practical performance we observed. For example, it is still unknown how Forgetting-MarI finds the unearthed baseline with certainty. Bridging this gap could lead to tighter analysis and potentially improved algorithms.

Second, our work highlights the importance of parameter selection in unlearning effectiveness. Developing principled approaches for optimal parameter tuning, especially with theoretical guidance, remains an open challenge that could significantly enhance the practicality of unlearning methods. Additionally, future work could explore the scalability of our approach to even larger models and datasets, investigate our method's robustness across different model architectures and domains, [replace retain set with new data set to remove retain access assumption or for more robust fine-tuning], and the principles of our approach could be applied to models trained on other data modalities.

As LLMs continue to grow in capability and deployment, developing reliable and theoretically grounded unlearning methods becomes increasingly important for responsible AI development and deployment. Forgetting-MarI is an important step towards that end. ‡

---

‡During the preparation of this work, the authors used large language model ChatGPT by OpenAI to refine the language and enhance readability. After using this tool or service, the authors reviewed and edited the content as needed and take full responsibility for the content of the publication.

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

CONTENTS

# A APPENDIX OF SECTION 1

## A.1 DETAILS OF LLM UNLEARNING METHODS: IMPLICIT MARGINAL INFORMATION UNLEARNING [VIA CONFLICTING FORCES]

Recent surveys highlight four broad families of LLM-unlearning techniques, each making a different compromise between *unlearn efficacy*, the ability to remove information from a model, *utility preservation*, how well the model performs on the remaining data, and *computational cost*, the resources expended to perform the unlearning [25]. A heuristic commonality of the techniques is their implicit/indirect target of marginal unlearning: all the methods tend to detect and thereby remove only the marginal effect of adding an "unlearn set" ($\mathcal{D}_u$), the dataset that is meant to be forgotten, to a "retain set" ($\mathcal{D}_r$), the dataset that the model should remember, on the given model.

**Full parameter fine-tuning:** These techniques train and perform weight updates on the whole model. Gradient ascent (or "loss reversal") [42] is the most straight-forward unlearning technique. It directly maximizes the cross-entropy on $\mathcal{D}_u$, effectively penalizing the model performance on the unlearn set. This type of unlearning was shown to lead to an overall decrease in model performance, so Gradient Difference [25] was developed to balance unlearning while maintaining general model performance. Gradient Difference maximizes the cross-entropy loss on $\mathcal{D}_u$ while continuing to *minimize* the loss on $\mathcal{D}_r$:

$$\min_\theta \underbrace{\mathbb{E}_{x\in\mathcal{D}_r}\ell(\theta;x)}_{\text{utility}} - \lambda \underbrace{\mathbb{E}_{x\in\mathcal{D}_u}\ell(\theta;x)}_{\text{loss reversal}},$$

$$\ell(\theta;x) = \mathrm{CE}\left(p_\theta(\,\cdot\mid x_{<t}), x_t\right).$$

Here $\lambda > 0$ balances utility preservation and unlearning. Intuitively, gradient descent is applied on $\mathcal{D}_r$ while gradient *ascent* is applied on $\mathcal{D}_u$.

Follow-up studies revealed that, even when balanced with gradient descent, this global ascent signal is too coarse: it suppresses the target examples but also degrades correlated yet legitimate content [17]. To overcome this challenge, variants have aimed to improve both sides of the problem. For utility preservation, past work has shown that distillation-style regularization with a Kullback–Leibler (KL) divergence penalty outperforms gradient descent on $\mathcal{D}_r$ in keeping the updated model close to the original without over-training on the retain set. For unlearning, alignment-style variants such as Direct Preference Optimization (DPO) and Negative Preference Optimization (NPO) replace the unlearn objective with more specific preference-based objectives, slowing catastrophic performance collapse [33, 43]. However, such preference-supervised methods can be difficult to generalize to unlearn at a large scale.

Finally, from the perspective of *marginal* unlearning, these full-parameter objectives act as *indirect proxies* for the marginal effect of adding $\mathcal{D}_u$ to $\mathcal{D}_r$: they rely on carefully balancing ascent on $\mathcal{D}_u$ and descent (or KL regularization) on $\mathcal{D}_r$. In practice, such proxies can be neither the most *effective* nor the most *efficient* at isolating the unique contribution of $\mathcal{D}_u$ without erasing information shared with $\mathcal{D}_r$.

**Weight editing and partial tuning:** In an effort to perform unlearning more efficiently, this line of methods focuses on selectively altering only a subset of a model's parameters rather than retraining the entire network. Such "model-surgery" methods perform rank-constrained updates at one or a few layers. Rank-One Model Editing (ROME) edits a single MLP weight with a closed-form rank-1 patch [30]. In particular, it modifies only the weights causally responsible for one token sequence in the unlearn set:

$$\min_{\Delta W} \|\Delta W\|_F^2 \ \text{ s.t. } \ W_{l^\star} h_{l^\star}(x) + \Delta W h_{l^\star}(x) = v_{\text{new}}.$$

Here, $\Delta W := \frac{\left(v_{\text{new}}-v_{\text{old}}\right)h^\top}{\|h\|_2^2}$, $l^*$ is the layer most influenced by the unlearn sample or prompt $x$, $h_{l^*}(x)$ is the activation and $W_{l^*}$ is the weight matrix of layer $l^*$, $v_{old} := W_{l^*}h_{l^*}(x)$, and finally $v_{new}$ is the alternative answer we want to replace $v_{old}$ by. Mass Editing Memory in a Transformer (MEMIT) [31] extends this idea to thousands of facts simultaneously and stacks many $(h_{l^*}^i, v_{l^*}^i)$ pairs. AlphaEdit furthers the idea by projecting edits into the null space of preserved knowledge, with the aim to improve robustness in sequential settings, ensuring minimal disruption to previously learned information. Detecting and Editing Privacy Neurons (DEPN) [39] masks the gradients of

neurons identified as contributing the most to the prediction of privacy-related content. In general, weight editing and partial tuning techniques are fast, but they are limited to short factual associations and struggle with stylistic or distributed knowledge.

Finally, the above weight-editing and partial-tuning methods share a common *indirect marginal unlearning* proxy: they infer marginal information by targeting parameters most influenced by the unlearn set, while largely ignoring parameters most influenced by the retain set. This can help isolate some marginal information signal, but again risks overlooking deep interactions between $r$ and $u$.

**Curating counterfactuals:** Instead of directly unlearning all or part of the model, another approach is to substitute the parametric knowledge of the unlearn set with benign knowledge. Broadly, this class of methods can be characterized by:

$$\min_\theta \underbrace{\mathbb{E}_{x \in \mathcal{D}_r}\big[\ell(\theta; x)\big]}_{\text{retain utility}} + \lambda \underbrace{\mathbb{E}_{x \in \mathcal{D}_{\text{neg}}}\big[\ell(\theta; x)\big]}_{\text{counterfactual prompts}},$$

where $\mathcal{D}_{\text{neg}}$ contains prompts or contexts designed to *neutralize* the influence of the unlearn set, $\ell(\theta; x)$ is the same cross entropy loss as before, and $\lambda > 0$ balances unlearning against utility.

"I don't know" [20] trains the model on question-answer pairs that map sensitive questions to a safe refusal (e.g. "I don't know"), teaching the model to decline queries about the unlearn set. Entity anonymization [9] replaces sensitive entities with anonymized placeholders and trains the model on the rewritten placeholders to scrub identifiable information from the model. Unlearning Large Language Models via Negative Response and Model Parameter Average (ULMR) [36] constructs adversarial "negative" prompts, trains on the paired responses, and then averages the updated weights with the base model to dampen overshoot. Selective Knowledge-negation Unlearning (SKU) first mines harmful or copyrighted contexts via red-teaming, then injects counterfactuals that negate them [26]. Such approaches are easy to deploy but depend heavily on prompt engineering and high-quality counterexamples.

From a marginal information proxy perspective, the curating counterfactuals approach aims to first penalize model utility related to the unlearn set by replacing the original model capability on the unlearn set with a lower-utility capacity on the counterfactuals, then rescue the utility related to the retain set using the utility preservation term, and finally balance the two to indirectly find the marginal information and penalize it.

**Model adaptation:** These methods train something external to the model and then use that externally trained adapter to update the model itself. A common instantiation is the task-vector framework: let $p_{\theta_0}$ be the original model and $p_{\theta_u}$ the same model fine-tuned on the unlearn set $\mathcal{D}_u$. The element-wise difference $\Delta\theta := \theta_u - \theta_0$ is treated as an encoding of the deleted knowledge and direct-subtraction methods [19] form the unlearned model as $p_{\theta_0 - \Delta\theta}$. Orthogonality offers an alternative geometric control. $O^3$ [12] trains one orthogonal LoRA adapter per removal request and learns a contrastive out-of-distribution (OOD) gate that activates the corresponding adapter at inference time. Orthogonality limits interference between requests, but the approach incurs two key costs: (1) the number of adapters (and hence memory) grows linearly with the number of unlearning requests, and (2) any mismatch between model behavior and the assumed linear/inner-product structure in weight space can undermine both unlearning guarantees and downstream utility.

From a *marginal-information* viewpoint, model-adaptation methods isolate the contribution of $\mathcal{D}_u$ by (i) subtracting the unlearn-induced task vector from the retain base, $\theta_0 - \Delta\theta$, or (ii) enforcing orthogonality between components aligned with retain and the unlearn signals and then penalize the isolated component. Both approaches can be considered as proxy of marginal information, though with strong arithmetic or geometric assumptions.

The proposed method, Forgetting-MarI, belongs to the full-parameter fine-tuning category. It applies a "marginal information" penalty that suppresses only the influence of the unlearn set while leaving the shared information, which is supported by the retain data, largely intact.

## B  APPENDIX OF SECTION 2

### B.1  PROOF OF PROPOSITION 2.1

*Proof.* To start, define the Bayes error as

$$P_e := \mathbb{E}_{X_{\mathrm{MarI}}}\Big[\min\big\{P(Z = 0 \mid X_{\mathrm{MarI}}),\, P(Z = 1 \mid X_{\mathrm{MarI}})\big\}\Big] = 1 - P_{acc}.$$

In addition, for each $x$, let $p(x) := P(Z = 1 \mid X_{\mathrm{MarI}} = x) \in [0, 1]$ be the conditional probability of $\{Z = 1\}$ given $\{X_{\mathrm{MarI}} = x\}$. Then it follows from $Z$ being binary that $H(Z \mid X_{\mathrm{MarI}} = x) = H_2\big(p(x)\big)$. Denote $m(X_{\mathrm{MarI}}) := \min\big\{P(Z = 0 \mid X_{\mathrm{MarI}}),\, P(Z = 1 \mid X_{\mathrm{MarI}})\big\}$. Since $H_2$ is concave, it follows from Jensen's inequality that

$$
\begin{aligned}
H(Z \mid X_{\mathrm{MarI}}) &= \mathbb{E}_{X_{\mathrm{MarI}}}\big[H_2(p(X_{\mathrm{MarI}}))\big] \\
&= \mathbb{E}_{X_{\mathrm{MarI}}}\big[H_2(m(X_{\mathrm{MarI}}))\big] \\
&\le H_2\big(\mathbb{E}_{X_{\mathrm{MarI}}}[m(X_{\mathrm{MarI}})]\big) \\
&= H_2(P_e).
\end{aligned}
$$

where the second equality holds due to the fact that $H_2(p) = H_2(1-p)$. Now, since $I(X_{\mathrm{MarI}}; Z) = H(Z) - H(Z \mid X_{\mathrm{MarI}})$ and $H(Z) = H_2(\pi)$, we obtain

$$H_2(\pi) - I(X_{\mathrm{MarI}}; Z) = H(Z \mid X_{\mathrm{MarI}}) \le H_2(P_e).$$

Since $P_e \in [0, \frac{1}{2}]$ and $H_2$ is strictly increasing on this interval, by applying the inverse $H_2^{-1}$, we have

$$P_e \ge H_2^{-1}\big(H_2(\pi) - I(X_{\mathrm{MarI}}; Z)\big).$$

Finally, by $P_{acc} = 1 - P_e$, we have

$$P_{acc} \le 1 - H_2^{-1}\big(H_2(\pi) - I(X_{\mathrm{MarI}}; Z)\big).$$

This proves the stated inequality. The particular case $\pi = \frac{1}{2}$ follows from $H_2(\frac{1}{2}) = 1$.

It remains to show that the upper bound is tight. Indeed, fix an arbitrary $I \in [0, H_2(\pi)]$. Choose $p^\star \in \big[\frac{1}{2}, 1\big]$ such that $H_2(p^\star) = H_2(\pi) - I$. Construct $P_{Z\mid X_{\mathrm{MarI}}}$ such that $P(Z = 1 \mid X_{\mathrm{MarI}}) \in \{p^\star, 1 - p^\star\}$ with probabilities chosen to match the prior $\pi$. Then $H(Z \mid X_{\mathrm{MarI}}) = H_2(p^\star)$ and $I(X_{\mathrm{MarI}}; Z) = I$, while the Bayes error satisfies $P_e = \min\{p^\star, 1 - p^\star\} = H_2^{-1}(H(Z) - I)$. Hence, equality holds in the bound. $\qquad\square$

### B.2  WHY MUTUAL INFORMATION RATHER THAN KL DIVERGENCE

One might consider penalizing a directional KL divergence between the "to-unlearn" and "to-retain" distributions. Instead, we regularize the *mutual information* between the model output and a binary indicator of sensitive content, which is equal to the Jensen-Shannon divergence as shown in Section 2. Here, we show that mutual information offers several advantages over one–way or two-way KL divergence:

- *Flexibility for utility and continual unlearning.* The reference $m$ in Jensen-Shannon divergence is the *mixture* of the two conditionals and evolves with training; we do not assume a fixed "gold" model. This yields a pure unlearning regularizer that can be combined with any utility term (e.g., $\ell_{\mathrm{KL}}(\theta, r)$) and naturally supports continual/online updates.
- *Stable training signal.* $I(\hat{X}; Z) \le H_2(\pi) \le \log 2$ for binary $Z$, so the gradients remain well-behaved even when supports differ, unlike one–way KL which can be unbounded on support mismatch.
- *Downstream robustness via data processing.* For any downstream representation or task $T = g(\hat{X})$, the data-processing inequality gives $I(T; Z) \le I(\hat{X}; Z)$. Thus, suppressing $I(\hat{X}; Z)$ at the model output (or an internal layer) upper-bounds leakage throughout the pipeline.

In contrast, a directional KL requires committing to a *fixed* target (encoding a specific utility assumption) and can be unstable or unbounded when supports are disjoint. That said, if an ideal frozen reference is indeed mandated, a one–way KL to that reference is a reasonable alternative.

## B.3 PROOF OF THEOREM 2.1

Here, we provide the proof for Theorem 2.1:

*Proof.* By the mean value theorem, for each $t$ there exists $\xi_t \in [\min\{p_t^u(u_t), p_t^r(u_t)\}, 1] \subseteq [\gamma, 1]$ such that

$$\left|\log p_t^u(u_t) - \log p_t^r(u_t)\right| = \frac{\left|p_t^u(u_t) - p_t^r(u_t)\right|}{\xi_t}$$

$$\leq \frac{\left|p_t^u(u_t) - p_t^r(u_t)\right|}{\gamma}$$

$$\leq \frac{\|p_t^u - p_t^r\|_1}{\gamma}$$

$$= \frac{2\|p_t^u - p_t^r\|_{TV}}{\gamma}.$$

Averaging over $t$,

$$\left|S_\theta(u, u) - S_\theta(u, r)\right| \leq \frac{2}{\gamma} \frac{1}{T} \sum_{t=1}^{T} \|p_t^u - p_t^r\|_{TV}.$$

Apply Lemma B.3 followed by Lemma B.2 and Jensen's inequality:

$$\frac{1}{T} \sum_t \|p_t^u - p_t^r\|_{TV} = \frac{1}{1-\alpha} \frac{1}{T} \sum_t \|p_t^d - p_t^r\|_{TV} \leq \frac{\sqrt{2}}{1-\alpha} \sqrt{\frac{1}{T} \sum_t \mathrm{JSD}(p_t^d, p_t^r)}.$$

Combining yields the claim. $\qquad\square$

## B.4 GENERALIZATION OF THEOREM 2.1 TO UNLEARN SET NEIGHBORHOOD

[Here, we show that the self-perplexity gap guarantee provided by Theorem 2.1 can be generalized to a neighborhood of $u$ rather than $u$ itself:]

**Theorem B.1** (MarI controls neighborhood-perplexity gap). *Draw $U := \{U_t\}_{t=1}^T$ with $U_t \sim p_t^u$ independently across $t \in [T]$ and suppose $[\max_{t,x} \left\{ \frac{p_t^u(x)}{p_t^r(x)} \vee \frac{p_t^r(x)}{p_t^u(x)} \right\} =: M < \infty]$. Let $C := \max_{t,x:\, p_t^u(x)>0} \left[\log \frac{p_t^r(x)}{p_t^u(x)}\right]^2 < \infty$. Then, for any $\varepsilon > 0$, with probability at least $1 - 2\exp\left(-T\varepsilon^2/(2C)\right)$,*

$$\left|S_\theta(U, u) - S_\theta(U, r)\right| \leq \left(\log M\right) \frac{M}{M-1} \frac{\sqrt{2}}{1-\alpha} \sqrt{I(X_{\mathrm{MarI}}; Z)} + \varepsilon.$$

We start with the following three lemmata that are needed for the proof of Theorem B.1:

**Lemma B.1** (Point-wise KL bound). *Let $p, q$ be two probability distributions over a finite set $V$ such that $\frac{p(x)}{q(x)} \in [1, M]$ for every $x \in V$ for some constant $M > 1$. Then for every $x \in V$*

$$p(x) \log \frac{p(x)}{q(x)} \leq (\log M) \frac{M}{M-1} \left[p(x) - q(x)\right]. \tag{4}$$

*Proof.* Fix $x \in V$ and set $y := \frac{p(x)}{q(x)} \in [1, M]$. Inequality equation 4 is equivalent to

$$y \log y \leq \frac{M}{M-1} (\log M)(y-1), \qquad \forall y \in [1, M]. \tag{4}$$

For $y > 1$ let $g(y) := \frac{y \log y}{y-1}$ and set $g(1) := \lim_{y \to 1^+} g(y) = 1$. We show that $g$ is strictly increasing on $[1, M]$. Indeed, compute $g'(y) = \frac{(y-1) - \log y}{(y-1)^2}$. Since $\log y < y - 1$ for all $y > 1$, we have $g'(y) > 0$; thus, $g$ is strictly increasing. Because $g$ is increasing and $y \in [1, M]$, we have

$$g(y) \leq g(M) = \frac{M \log M}{M-1}.$$

Multiplying both sides by $y - 1$ yields equation 4, which is precisely equation 4 after reinstating $y = p(x)/q(x)$. Therefore, equation 4 holds for every $x \in V$. This completes the proof. $\qquad \square$

**Lemma B.2. (Total Variation is controlled by Jensen-Shannon Divergence)** *For any two probability measures $p, q$ on a finite set, we have*

$$\|p - q\|_{TV} \leq \sqrt{2\,\mathrm{JSD}(p, q)},$$

*where* $\mathrm{JSD}(p, q) := \frac{1}{2} D_{\mathrm{KL}}(p\|m) + \frac{1}{2} D_{\mathrm{KL}}(q\|m)$, $m := \frac{p+q}{2}$ *and* $D_{\mathrm{KL}}(p\|q) := \sum_v p(v) \log \frac{p(v)}{q(v)}$, *denotes the Jensen–Shannon divergence.*

*Proof.* Let $m = \frac{p+q}{2}$. Pinsker's inequality gives $\|p - m\|_1^2 \leq 2\,D_{\mathrm{KL}}(p\|m)$ and analogously for $q$. Hence

$$\mathrm{JSD}(p, q) \geq \frac{1}{4}\Big[\|p - m\|_1^2 + \|q - m\|_1^2\Big] = \frac{1}{8}\|p - q\|_1^2,$$

because $p - m = \frac{p-q}{2}$ and $q - m = -\frac{p-q}{2}$. Since $\|p - q\|_{TV} = \frac{1}{2}\|p - q\|_1$, it follows that $\|p - q\|_{TV}^2 \leq 2\,\mathrm{JSD}(p, q)$. $\qquad \square$

**Lemma B.3** (Exact TV scaling under mixture)**.** *If $p^d = \alpha p^r + (1 - \alpha)p^u$ with $\alpha \in (0, 1)$, then*

$$\|p^u - p^r\|_{TV} = \frac{1}{1 - \alpha}\|p^d - p^r\|_{TV}.$$

*Proof.* $p^d - p^r = (1 - \alpha)(p^u - p^r)$. Taking $\ell_1$-norms and dividing by 2 yields the identity. $\qquad \square$

Now, we are ready to prove Theorem B.1:

*Proof.* Define $Y_t := \log \frac{p_t^r(U_t)}{p_t^u(U_t)}$, so that

$$S_\theta(U, u) - S_\theta(U, r) = \frac{1}{T}\sum_{t=1}^{T} Y_t.$$

Since $U_t \sim p_t^u$, $\mathbb{E}[Y_t] = \sum_x p_t^u(x) \log \frac{p_t^r(x)}{p_t^u(x)} = -D_{\mathrm{KL}}(p_t^u\|p_t^r)$, hence

$$\mathbb{E}\big[S_\theta(U, u) - S_\theta(U, r)\big] = -\frac{1}{T}\sum_{t=1}^{T} D_{\mathrm{KL}}(p_t^u\|p_t^r).$$

Now, by the assumption $\max_{t,x} \max\Big\{\frac{p_t^u(x)}{p_t^r(x)}, \frac{p_t^r(x)}{p_t^u(x)}\Big\} \leq M$, we have $p_t^r(x) > 0$ for $p_t^u(x)$-a.e. $x$ for all $t$. Therefore, for all $t$, we have $\log \frac{p_t^r(x)}{p_t^u(x)} < \infty$ and taking the maximum over $t \in [T]$, we obtain $C := \max_{t,\, x:\, p_t^u(x)>0} \Big[\log \frac{p_t^r(x)}{p_t^u(x)}\Big]^2 < \infty$. It then follows from the definition of $Y_t$ that $|Y_t| \leq \sqrt{C}$ a.s.. Hoeffding's inequality for independent bounded variables yields, for any $\varepsilon > 0$,

$$\mathbb{P}\left(\left|\frac{1}{T}\sum_{t=1}^{T} Y_t - \mathbb{E}\frac{1}{T}\sum_{t=1}^{T} Y_t\right| \geq \varepsilon\right) \leq 2\exp\left(-\frac{T\varepsilon^2}{2C}\right).$$

Using $\big||a| - b\big| \leq |a - b|$ for $b \geq 0$, we have

$$\big|S_\theta(U, u) - S_\theta(U, r)\big| \leq \frac{1}{T}\sum_{t=1}^{T} D_{\mathrm{KL}}(p_t^u\|p_t^r) + \varepsilon.$$

with probability at least $1 - 2\exp\big(-\frac{T\varepsilon^2}{2C}\big)$.

Now, for each $t$, let $A_t = \{x : p_t^u(x) \geq p_t^r(x)\}$. Then by Lemma B.1, we have

$$D_{\mathrm{KL}}(p_t^u\|p_t^r) \leq \kappa(M)\sum_{x \in A_t}\big(p_t^u(x) - p_t^r(x)\big) \leq \kappa(M)\|p_t^u - p_t^r\|_{TV}.$$

Averaging in $t$ gives

$$\frac{1}{T}\sum_{t=1}^{T} D_{\mathrm{KL}}(p_t^u \| p_t^r) \ \leq \ \kappa(M)\frac{1}{T}\sum_{t=1}^{T}\|p_t^u - p_t^r\|_{TV}.$$

Finally, it follows from Lemma B.3 and Lemma B.2 that

$$\frac{1}{T}\sum_{t=1}^{T}\|p_t^u - p_t^r\|_{TV} \ = \ \frac{1}{1-\alpha}\frac{1}{T}\sum_{t=1}^{T}\|p_t^d - p_t^r\|_{TV} \ \leq \ \frac{\sqrt{2}}{1-\alpha}\frac{1}{T}\sum_{t=1}^{T}\sqrt{\mathrm{JSD}(p_t^d, p_t^r)}.$$

By Jensen's inequality, $\frac{1}{T}\sum_t \sqrt{\mathrm{JSD}(p_t^d, p_t^r)} \ \leq \ \sqrt{\frac{1}{T}\sum_t \mathrm{JSD}(p_t^d, p_t^r)}$. Combining the displays proves the claim with $I(X_{\mathrm{MarI}}; Z) = \frac{1}{T}\sum_t \mathrm{JSD}(p_t^d, p_t^r)$. $\qquad\square$

# C    APPENDIX OF SECTION 3

## C.1    THEORETICAL ERROR BOUND BETWEEN POSITION-WISE VS. POOLED MARI

Here, we provide the theoretical analysis of the error between MarI and pooled MarI. The following result shows that, under a mild assumption, the error of using the pooled MarI to estimate MarI is bounded by the sequence-wise density variance:

**Theorem C.1** (Pooling error bound). *For each $t$, set $m_t := \frac{1}{2}(p_t^d + p_t^r)$ and $\bar{m} := \frac{1}{2}(\bar{p}^d + \bar{p}^r)$, where $\bar{p}^d := \frac{1}{T}\sum_{t=1}^{T} p_t^d$ and $\bar{p}^r := \frac{1}{T}\sum_{t=1}^{T} p_t^r$. Assume the uniform overlap condition*

$$\beta := \min\left\{ \inf_{\lambda \in [0,1]} \min_{t,x}\big[(1-\lambda)\,m_t(x) + \lambda\,\bar{m}(x)\big], \right. \tag{5}$$

$$\left. \inf_{\lambda \in [0,1]} \min_{t,x}\big[(1-\lambda)\,p_t^d(x) + \lambda\,\bar{p}^d(x)\big], \inf_{\lambda \in [0,1]} \min_{t,x}\big[(1-\lambda)\,p_t^r(x) + \lambda\,\bar{p}^r(x)\big]\right\} > 0. \tag{6}$$

*Define the (averaged) $\ell_2$-deviation terms*

$$V_d := \frac{1}{T}\sum_{t=1}^{T}\big\|p_t^d - \bar{p}^d\big\|_2^2, \qquad V_r := \frac{1}{T}\sum_{t=1}^{T}\big\|p_t^r - \bar{p}^r\big\|_2^2.$$

*Then*

$$0 \le I(X_{\mathrm{MarI}}; Z) - I(\bar{X}_{\mathrm{MarI}}; Z) \le \frac{1}{4\beta}\Big(V_d + V_r\Big), \tag{7}$$

*where $I(X_{\mathrm{MarI}}; Z) = \frac{1}{T}\sum_{t=1}^{T} \mathrm{JSD}(p_t^d, p_t^r)$ and $I(\bar{X}_{\mathrm{MarI}}; Z) = \mathrm{JSD}(\bar{p}^d, \bar{p}^r)$.*

*Proof.* The lower bound $I(\bar{X}_{\mathrm{MarI}}; Z) \le I(X_{\mathrm{MarI}}; Z)$ follows directly from the data-processing inequality. For the upper bound, write

$$\frac{1}{T}\sum_{t=1}^{T} \mathrm{JSD}(p_t^d, p_t^r) - \mathrm{JSD}(\bar{p}^d, \bar{p}^r)$$

$$= \frac{1}{T}\sum_{t=1}^{T}\Big[ \underbrace{H(m_t) - H(\bar{m})}_{(A_t)} - \frac{1}{2}\underbrace{\big(H(p_t^d) - H(\bar{p}^d)\big)}_{(B_t)} - \frac{1}{2}\underbrace{\big(H(p_t^r) - H(\bar{p}^r)\big)}_{(C_t)}\Big].$$

Let $\Delta_m := m_t - \bar{m}$, $\Delta_d := p_t^d - \bar{p}^d$, $\Delta_r := p_t^r - \bar{p}^r$. Since $H$ is twice-differentiable, the second-order Taylor expansion around the pooled densities yields

$$H(a) = H(a') + \langle \nabla H(a'), a - a' \rangle + \tfrac{1}{2}(a - a')^\top \nabla^2 H(s)(a - a'),$$

for some $s$ on the line segment between $a'$ and $a$. Note that $H$ is concave, $\nabla^2 H(s)$ is negative semidefinite and diagonal with entries $-1/s(x)$. By the overlap assumption equation 5, every coordinate along the segments between $m_t$ and $\bar{m}$ and between $p_t^s$ and $\bar{p}^s$ ($s \in \{d, r\}$) is at least $\beta$, hence

$$-(a - a')^\top \nabla^2 H(s)(a - a') \le \frac{1}{\beta}\sum_{x \in V}\big(a(x) - a'(x)\big)^2 \le \frac{1}{\beta}\|a - a'\|_2^2,$$

and therefore the (negative) Taylor remainders satisfy

$$H(a) - H(a') - \langle \nabla H(a'), a - a' \rangle \ge -\frac{1}{2\beta}\|a - a'\|_2^2. \tag{8}$$

Applying equation 8 to the three entropy differences and averaging in $t$, the first-order terms vanish because $\frac{1}{T}\sum_t \Delta_d = 0$, $\frac{1}{T}\sum_t \Delta_r = 0$, and $\frac{1}{T}\sum_t \Delta_m = 0$. Thus,

$$
\begin{aligned}
\frac{1}{T}\sum_t \left[(A_t) - (B_t)/2 - (C_t)/2\right] &\leq \frac{1}{T}\sum_t \frac{1}{2}\Big(-\mathcal{R}_d(t)\Big) + \frac{1}{T}\sum_t \frac{1}{2}\Big(-\mathcal{R}_r(t)\Big) - \frac{1}{T}\sum_t \mathcal{R}_m(t)\\
&\leq -\frac{1}{2}\Big(\frac{1}{T}\sum_t \big(\mathcal{R}_d(t)\big) + \frac{1}{T}\sum_t \big(\mathcal{R}_r(t)\big)\Big)\\
&\leq \frac{1}{4\beta}\Big(\frac{1}{T}\sum_t \big(\|p_t^d - \bar{p}^d\|_2^2\big) + \frac{1}{T}\sum_t \big(\|p_t^r - \bar{p}^r\|_2^2\big)\Big)\\
&= \frac{1}{4\beta}\left(V_d + V_r\right).
\end{aligned}
$$

where $\mathcal{R}_s(t) := H(p_t^s) - H(\bar{p}^s) - \langle \nabla H(\bar{p}^s), p_t^s - \bar{p}^s\rangle \leq 0$ for $s \in \{d, r\}$ and $\mathcal{R}_m(t) := H(m_t) - H(\bar{m}) - \langle \nabla H(\bar{m}), m_t - \bar{m}\rangle \leq 0$. $\qquad\square$

## C.2 THEORETICAL GUARANTEE PROVIDED BY POOLED MARI

In general, there is *no non-trivial (distribution–free)* theoretical guarantee for the sequence–level perplexity gaps can be stated solely in terms of the pooled MarI $I(\bar{X}_{\mathrm{MarI}}; Z) = \mathrm{JSD}(\bar{p}^d, \bar{p}^r)$ without any additional assumption, such as the bounded $\ell^2$-deviation in Theorem C.1 above.

Nonetheless, the pooled MarI *can* certify the following *word–level forgetting* (for particular tokens) forgetting:

**Theorem C.2** (Word-level provable unlearning via pooled MarI). *Fix a token $w \in V$ and assume $\bar{p}^u(w) \wedge \bar{p}^r(w) =: \bar{\gamma}_w \in (0, 1]$. Then*

$$
\Big| \log \bar{p}^u(w) - \log \bar{p}^r(w) \Big| \leq \frac{2}{\bar{\gamma}_w\,(1-\alpha)}\,\sqrt{2\,I(\bar{X}_{\mathrm{MarI}}; Z)}.
$$

*Proof.* It follows from mean value theorem for $x \mapsto \log x$ on $[\bar{\gamma}_w, 1]$ that

$$
|\log \bar{p}^u(w) - \log \bar{p}^r(w)| \leq |\bar{p}^u(w) - \bar{p}^r(w)|/\bar{\gamma}_w.
$$

Since $\bar{p}^u - \bar{p}^r = (\bar{p}^d - \bar{p}^r)/(1-\alpha)$, we have

$$
|\bar{p}^u(w) - \bar{p}^r(w)| \leq \|\bar{p}^u - \bar{p}^r\|_1 = \frac{1}{1-\alpha}\,\|\bar{p}^d - \bar{p}^r\|_1 = \frac{2}{1-\alpha}\,\|\bar{p}^d - \bar{p}^r\|_{TV}.
$$

Finally, combining the above two inequalities, we have

$$
|\log \bar{p}^u(w) - \log \bar{p}^r(w)| \leq \frac{2}{\bar{\gamma}_w(1-\alpha)}\,\|\bar{p}^d - \bar{p}^r\|_{TV} \leq \frac{2}{\bar{\gamma}_w(1-\alpha)}\,\sqrt{2\,I(\bar{X}_{\mathrm{MarI}}; Z)},
$$

where the last inequality follows from the fact that

$$
\|\bar{p}^d - \bar{p}^r\|_{TV} \leq \sqrt{2\,\mathrm{JSD}(\bar{p}^d, \bar{p}^r)} = \sqrt{2\,I(\bar{X}_{\mathrm{MarI}}; Z)}.
$$

$\qquad\square$

## C.3 EMPIRICAL ERROR BOUND BETWEEN POSITION-WISE VS. POOLED MARI

We empirically compare the token/position-wise MarI, $I(X_{\mathrm{MarI}}; Z) = \frac{1}{T}\sum_{t=1}^{T} I(X_t; Z)$, with the pooled ("flattened") MarI, $I(\bar{X}_{\mathrm{MarI}}; Z)$, on our heterogeneous dataset. As predicted by the data–processing inequality, $I(\bar{X}_{\mathrm{MarI}}; Z) \leq I(X_{\mathrm{MarI}}; Z)$, so the position-wise estimator produces a stronger marginal-information signal. Nevertheless, by appropriately tuning the trade-off parameter $\gamma$ (weighting MarI vs. utility), both estimators attain comparable forget–utility trade-offs.

However, we can also observe the influence of the heterogeneity of dataset and random batch sampling. In particular, in Figure 6, the position-wise estimator exhibits higher variance on heterogeneous batches (varying lengths, topics, and token alignments). Furthermore, Figure 7 shows that,

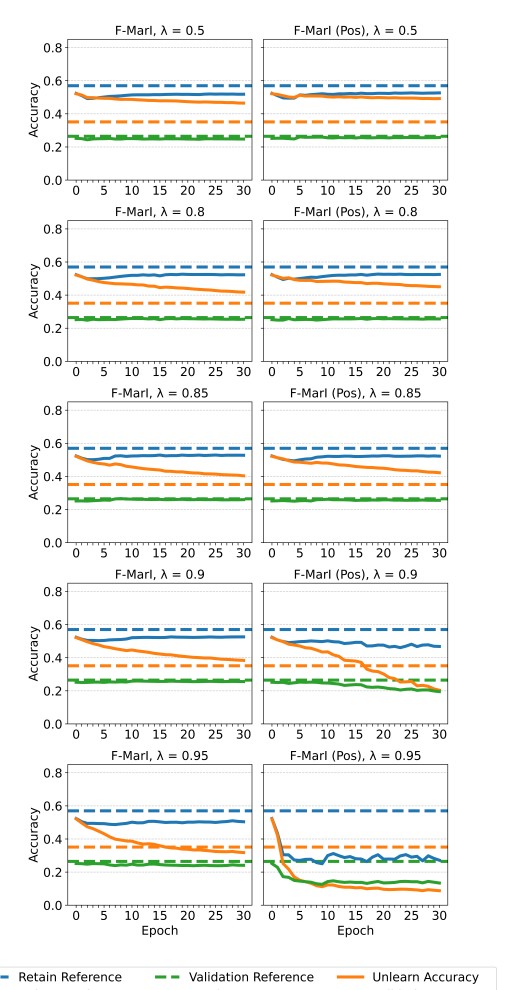

Figure 6: Position-wise vs. pooled MarI under several λ settings using Llama models.

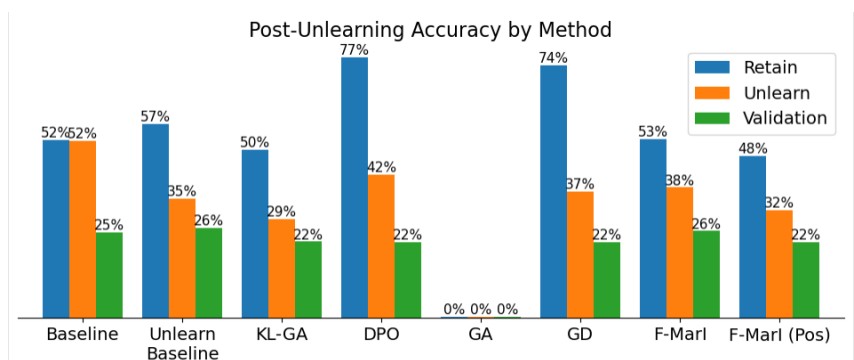

Figure 7: With a fixed trade-off ($\lambda = 0.9$) on LLama models, position-wise MarI is noisier on heterogeneous data and over-unlearns compared to the unlearn baseline.

with fixed $\gamma$ (e.g., $\gamma = 0.9$), the position-wise MarI tends to over-unlearn relative to the gold unlearn baseline. Intuitively, it can over-penalize idiosyncratic, position-specific fluctuations rather than true marginal effects.

In our experiments, because the text is heterogeneous in both length and context, we use random mini-batches and the pooled estimator by default.

# D APPENDIX OF SECTION 4

## D.1 GPU, COMPUTATION AND THE ALGORITHM

For the GPT2-LG model, all experiments use 4×NVIDIA A100-40GB GPUs in fp32 precision. For all unlearning methods, we use a per-GPU batch size of 8 for both the unlearn and retain sets, yielding an effective batch size of 32 examples for each set per optimization step.

| Method | Time per batch (s / batch) | Peak memory (GB / GPU) |
|--------|---------------------------|------------------------|
| F-MarI | 0.56 | 34.19 |
| KL-GA  | 0.50 | 31.22 |
| GD     | 0.45 | 27.15 |
| DPO    | 0.57 | 36.35 |
| GA     | 0.32 | 28.42 |

Table 4: Compute cost for different unlearning methods on GPT-2 Large. We report the average time per batch, and peak per-GPU memory usage.

During unlearning of the CP using Llama, the batch size was 3 for the unlearn set and 6 for the retain set. All experiments were conducted using 2 × NVIDIA GRID T4-16Q 16GB GPUs at maximum memory utilization.

| Method | Time per batch (s / batch) | Peak memory (GB / GPU) |
|--------|---------------------------|------------------------|
| F-MarI | 1.95 | $\sim 16$ |
| KL-GA  | 1.93 | $\sim 16$ |
| GD     | 1.50 | $\sim 16$ |
| DPO    | 1.76 | $\sim 16$ |
| GA     | 0.63 | $\sim 16$ |

## D.2 FULL ALGORITHM AND FLOW CHART

Here, we first provide both the full-detailed algorithm pseudo-code for Forgetting-MarI and the flowchart for readers who are more familiar with chart presentations.

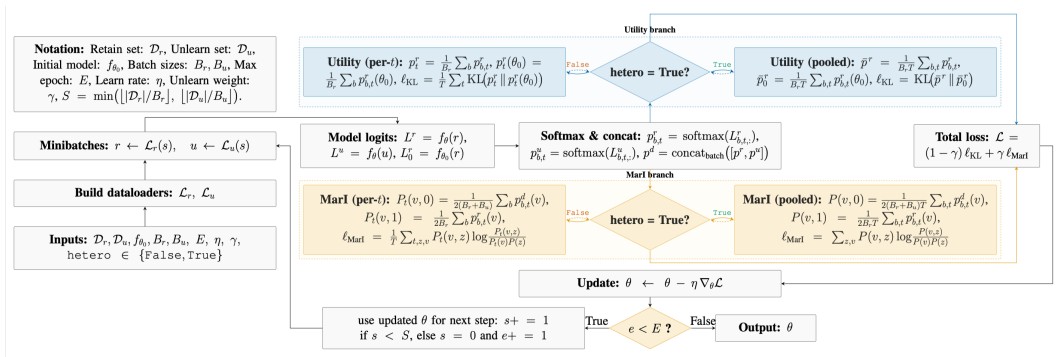

Figure 8: Flow chart for Forgetting-MarI.

Algorithm 1: Forgetting-MarI.

---

**Algorithm 2:** Forgetting-MarI

**Require:** Retain dataset $\mathcal{D}_r = \{x_i^r\}_{i=1}^{|\mathcal{D}_r|}$, unlearn dataset $\mathcal{D}_u = \{x_j^u\}_{j=1}^{|\mathcal{D}_u|}$; initial model $f_{\theta_0}$; learning rate $\eta$; batch sizes $B_r, B_u$; epochs $E$; trade-off $\gamma \in (0, 1)$; hetero $\in \{\text{False}, \text{True}\}$.

**Ensure:** Unlearned parameters $\theta_E$.

1: Initialize $\theta \leftarrow \theta_0$; build dataloaders $\mathcal{L}_r, \mathcal{L}_u$.
2: **for** $e = 1$ **to** $E$ **do**
3:    Shuffle $\mathcal{L}_r, \mathcal{L}_u$; $S \leftarrow \min\{|\mathcal{L}_r|, |\mathcal{L}_u|\}$.
4:    **for** $s = 1$ **to** $S$ **do**
5:       **Minibatches:** $r \leftarrow \mathcal{L}_r(s), \quad u \leftarrow \mathcal{L}_u(s)$.
6:       **Model logits & probabilities:**
7:          $L^r \leftarrow f_\theta(r) \in \mathbb{R}^{B_r \times T \times |V|}$, $p_{b,t}^r \leftarrow \text{softmax}(L^r[b, t, :]) \in [0, 1]^{B_r \times T \times |V|}$;
8:          $L_0^r \leftarrow f_{\theta_0}(r) \in \mathbb{R}^{B_r \times T \times |V|}$, $p_{b,t}^r(\theta_0) \leftarrow \text{softmax}(L_0^r[b, t, :]) \in [0, 1]^{B_r \times T \times |V|}$;
9:          $L^u \leftarrow f_\theta(u) \in \mathbb{R}^{B_u \times T \times |V|}$, $p_{b,t}^u \leftarrow \text{softmax}(L^u[b, t, :]) \in [0, 1]^{B_u \times T \times |V|}$;
10:        $p^d \leftarrow \text{concat}_{\text{batch}}([p^r, p^u]) \in [0, 1]^{(B_r + B_u) \times T \times |V|}$.
11:       **Utility Loss:**
12:       **If** hetero=True:
13:          $\bar{p}^r \leftarrow \frac{1}{B_r * T} \sum_{b,t} p_{b,t}^r$, $\bar{p}_0^r \leftarrow \frac{1}{B_r * T} \sum_{b,t} p_{b,t}^r(\theta_0)$,
14:          $\ell_{\text{KL}} \leftarrow \sum_{v \in V} \bar{p}^r(v) \log(\frac{\bar{p}^r(v)}{\bar{p}_0^r(v)})$
15:       **Else:**
16:          $p_t^r \leftarrow \frac{1}{B_r} \sum_{b=1}^{B_r} p_{b,t}^r$, $p_t^r(\theta_0) \leftarrow \frac{1}{N_r} \sum_{b=1}^{B_r} p_{b,t}^r(\theta_0)$,
17:          $\ell_{\text{KL}} \leftarrow \frac{1}{T} \sum_t \left( \sum_{v \in V} p_t^r(v) \log(\frac{p_t^r(v)}{p_t^r(\theta_0)(v)}) \right)$
18:       **MarI Loss:**
19:       **If** hetero=True:
20:          $P(v, 0) \leftarrow \frac{1}{2(B_r + B_u) * T} \sum_{b,t} p_{b,t}^d$, $P(v, 1) \leftarrow \frac{1}{2 B_r * T} \sum_{b,t} p_{b,t}^r$,
         $P(v) = \sum_{z \in \{0,1\}} P(v, z)$,
21:          $\ell_{\text{MarI}} \leftarrow \sum_{z \in \{0,1\}} \sum_{v \in V} P(v, z) \log \frac{P(v,z)}{P(v) P(z)}$.
22:       **Else:**
23:          $P_t(v, 0) \leftarrow \frac{1}{2(B_r + B_u)} \sum_b p_{b,t}^d$, $P_t(v, 1) \leftarrow \frac{1}{2 B_r} \sum_b p_{b,t}^r$, $P_t(v) = \sum_z P_t(v, z)$,
24:          $\ell_{\text{MarI}} \leftarrow \frac{1}{T} \sum_t \left( \sum_{z \in \{0,1\}} \sum_{v \in V} P_t(v, z) \log \frac{P_t(v,z)}{P_t(v) P_t(z)} \right)$.
25:       **Total loss:** $\mathcal{L} = (1 - \gamma) \ell_{\text{KL}} + \gamma \ell_{\text{MarI}}^{\text{flat}}$.
26:       **Update:** $\theta \leftarrow \theta - \eta \nabla_\theta \mathcal{L}$.
27:    **end for**
28: **end for**
29: **return** $\theta_E \leftarrow \theta$

---

Here, we note that, in practice, encoding often introduces padding tokens, and one should ignore those for downstream calculations: $X_R^{\text{flat}} \leftarrow \text{flatten}(L_R[i : x_i \neq \text{pad}]) \in \mathbb{R}^{N_r \times V}$; and do similarly for $X_U^{\text{flat}}$. This ensures that probabilities derived from $L_R^{\text{flat}}$ are not biased by padding positions.

## D.3   ABLATION STUDY FOR THE GPT2-LG

In this ablation study, we deliberately curated the datasets to be maximally correlated, creating conditions where the distinction between forget and retain information is most difficult. This setup serves as a stress test for unlearning methods, as it requires the algorithm to selectively remove knowledge that is deeply intertwined with information that should be preserved.

**Different** $\gamma$ Figure 10 reports the training curves of all the compared full-parameter tuning UL methods using different regularization parameters $\gamma$. Due to the marginal information unlearning nature, Forgetting-MarI has the advantage that a wide range of parameter choice results in fast convergence around the unlearn baseline, indicating robust parameter-tuning. In comparison, it is

clear that other methods demonstrate non-convergent (unstable) learning trajectory with extremely narrow parameter range to get close to the unlearn baseline, indicating extreme difficulty and effort in parameter-tuning.

**Learning Rate** Figure 9 tests a larger learning rate scenario among all methods.

These results collectively demonstrate that Forgetting-MarI provides more precise control over the unlearning process, maintaining the critical balance between forgetting targeted information and preserving general utility. The results highlights two notions of robustness:

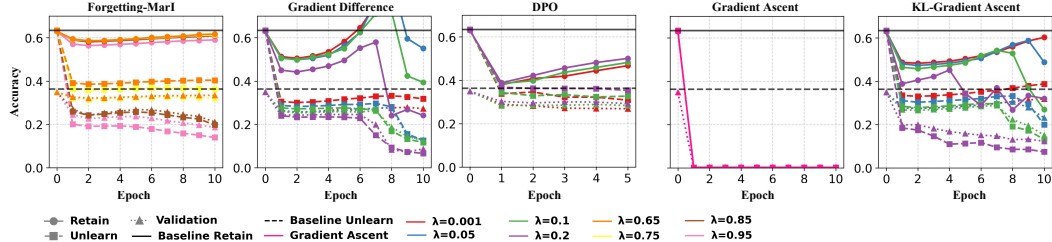

Figure 9: Training curves for each method with varying choices of the regularization parameter $\gamma$ and lr=1e-4. Forgetting-MarI exhibits smooth monotone behavior, while the other methods show oscillation or utility collapse.

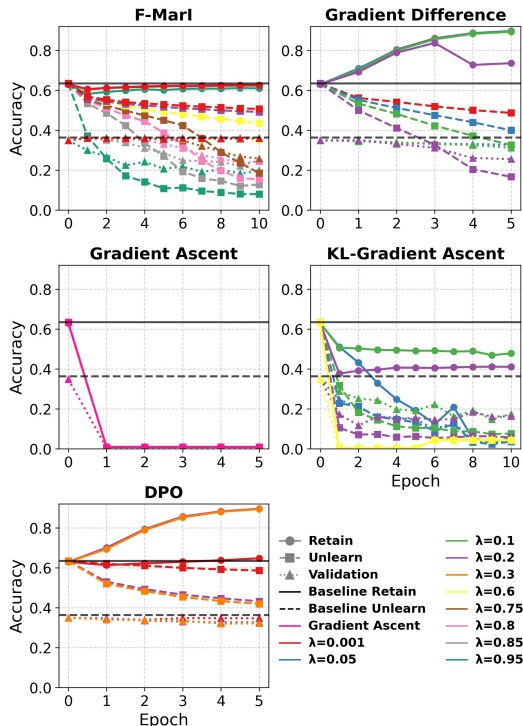

Figure 10: Training Curves for full-parameter UL methods with different $\gamma$ choices with lr= 1e-5.

(1) *Training robustness over epochs.* Forgetting-MarI descends steadily to its optimum as seen in Fig. 9. GA and GD overshoot and bounce, KL–GA diverges after 5–6 epochs, DPO plateaus prematurely.

(2) *Robustness against regularization tuning.* Forgetting-MarI shows a monotone and smooth utility-unlearning trade-off when adjusting the regularization parameter. In comparison, GD and KL-GA display unstable oscillations with different choices of $\gamma$.

Both training and regularization robustness are necessary for a practical use of unlearning techniques. Practitioners do not have access to ground truth baselines and have limited time to select/determine the best parameter or training epoch to stop at, so stability is essential. Forgetting-MarI is the most stable technique during unlearning, making it the safest choice in practice.

## D.4 SUPPLEMENTAL TO SEC 4.2

**Stability of continual unlearning** Figure 11 below shows the unlearning trajectories of different methods on GPT-2 with Harry Porter and Llama 3.2-1B with Careless People. The unlearning performance of each method over the course of unlearning, where the curves for each method correspond to the experiment with the best performing regularization parameter for each method.

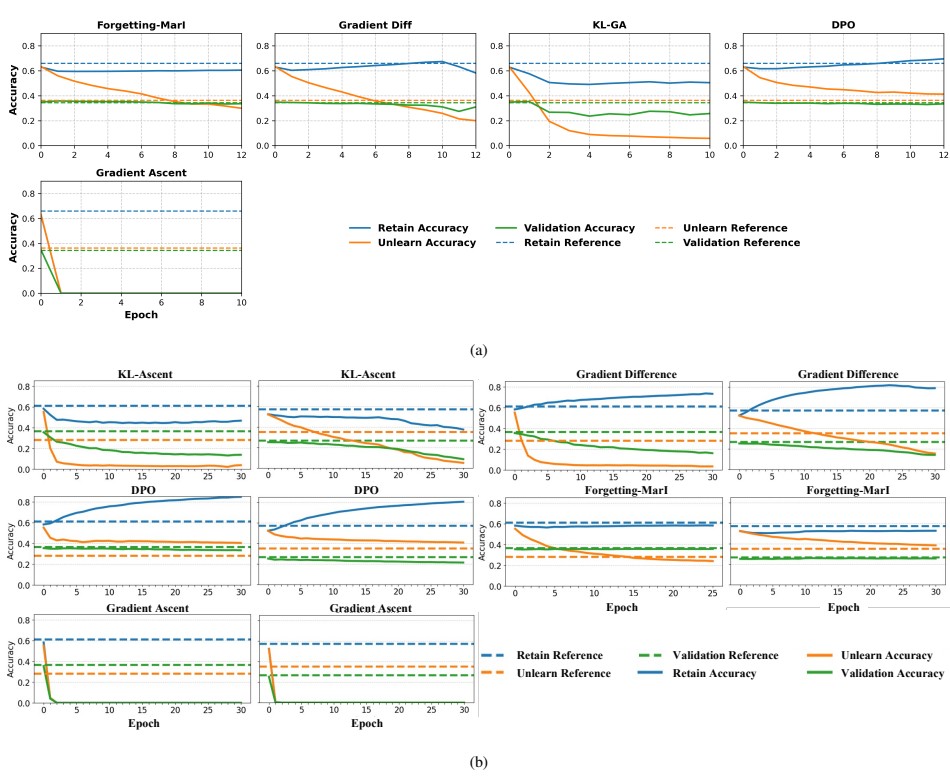

Figure 11: Next-token prediction accuracy during unlearning training across different methods. Horizontal dashed lines represent the "gold standard" unlearn baseline (model trained only on the retain dataset $\mathcal{D}_r$). **(a) Top:** Results for GPT-2 Large on Harry Potter, showing training dynamics across epochs (epoch 0 represents pre-unlearning performance). **(b) Bottom:** Results for Llama-3.2-1B on Careless People, with left and right columns for each method showing correlated and uncorrelated test settings, respectively.

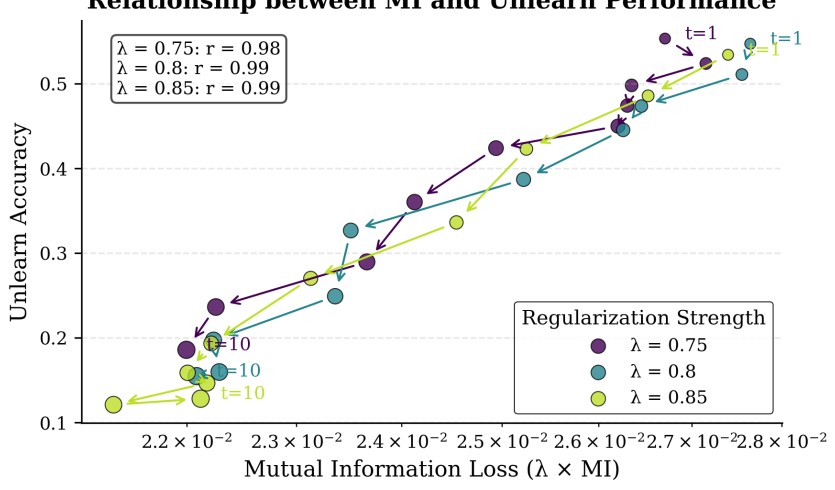

Figure 12: MI Loss for GPT-2 Large on Harry Potter Dataset. Here, we observe nearly perfect correlation between the designed marginal information regularization loss and the unlearn accuracy, under different choice of regularization parameter.)

Forgetting-MarI is the best at smoothly approximating the unlearn baseline. The methods based on gradient ascent, GA, GD, and KL-GA, all over-penalize $\mathcal{D}_u$ due to the utility-destroying nature of gradient ascent. DPO, meanwhile, never matches the unlearn baseline in accuracy on $\mathcal{D}_u$ and over-trains on $\mathcal{D}_r$. Across both sets of experiments, Forgetting-MarI minimally affects the validation accuracy, as seen by the validation curve remaining largely unchanged.

We note that there is the possibility that, in theory, one could find a perfect balance between gradient ascent and utility regularization, leading to a stable balance between unlearning and utility preservation, using one of the other methods. However, such a balance seems practically unattainable due to the unlearning instability over time and the lack of monotonicity in the choice of $\gamma$ for methods based on gradient ascent.

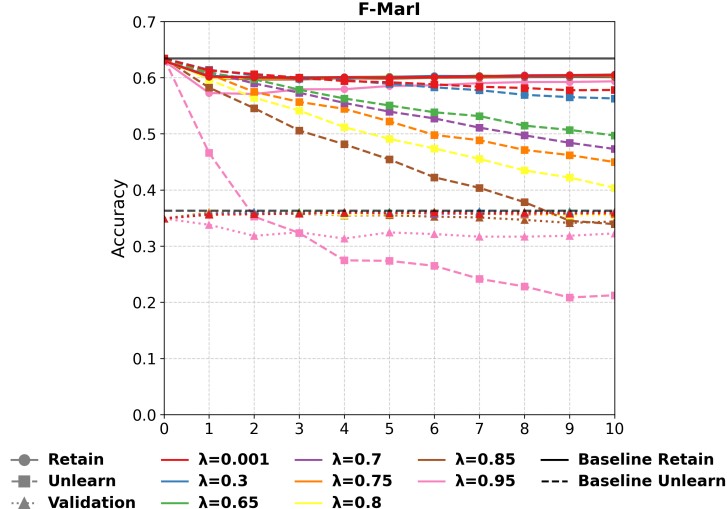

Figure 13: Training curves of F-MarI for the 10/90 split unlearning with the GPT2-LG model.

Additionally, we report the training curves for various $\gamma$ for F-MarI for the HP in Figure 13. We also plot the proposed MI loss across a few $\gamma$.

Finally, in Figure 12, we report an observation of nearly perfect correlation between the designed marginal information regularization loss and the unlearn accuracy, under different choice of regularization parameter.

### D.5 SUPPLEMENTARY GENERAL MODEL CAPACITY TEST RESULTS

Table 5 and 6 summarize comprehensive evaluation results across multiple benchmark tests.

| Task | Metric | GPT2-LG | Baseline | Unlearn Baseline | F-MarI | KL-GA | GA | GD | DPO |
|------|--------|---------|----------|------------------|--------|-------|-----|-----|-----|
| ARC-Easy | acc | 0.53 ± 0.01 | 0.46 ± 0.01 | 0.48 ± 0.01 | 0.46 ± 0.01 | 0.46 ± 0.01 | 0.24 ± 0.01 | 0.45 ± 0.01 | 0.46 ± 0.01 |
|  | acc_norm | 0.47 ± 0.01 | 0.42 ± 0.01 | 0.43 ± 0.01 | 0.43 ± 0.01 | 0.43 ± 0.01 | 0.25 ± 0.01 | 0.43 ± 0.01 | 0.43 ± 0.01 |
| ARC-Challenge | acc | 0.22 ± 0.01 | 0.23 ± 0.01 | 0.22 ± 0.01 | 0.23 ± 0.01 | 0.24 ± 0.01 | 0.22 ± 0.01 | 0.23 ± 0.01 | 0.23 ± 0.01 |
|  | acc_norm | 0.25 ± 0.01 | 0.27 ± 0.01 | 0.27 ± 0.01 | 0.27 ± 0.01 | 0.28 ± 0.01 | 0.26 ± 0.01 | 0.27 ± 0.01 | 0.27 ± 0.01 |
| PIQA | acc | 0.70 ± 0.01 | 0.66 ± 0.01 | 0.66 ± 0.01 | 0.66 ± 0.01 | 0.66 ± 0.01 | 0.53 ± 0.01 | 0.65 ± 0.01 | 0.65 ± 0.01 |
|  | acc_norm | 0.69 ± 0.01 | 0.65 ± 0.01 | 0.65 ± 0.01 | 0.66 ± 0.01 | 0.66 ± 0.01 | 0.51 ± 0.01 | 0.64 ± 0.01 | 0.65 ± 0.01 |
| Hellaswag | acc | 0.36 ± 0.00 | 0.36 ± 0.00 | 0.36 ± 0.00 | 0.35 ± 0.00 | 0.36 ± 0.00 | 0.25 ± 0.00 | 0.35 ± 0.00 | 0.36 ± 0.00 |
|  | acc_norm | 0.45 ± 0.00 | 0.43 ± 0.00 | 0.43 ± 0.00 | 0.42 ± 0.00 | 0.42 ± 0.00 | 0.26 ± 0.00 | 0.42 ± 0.00 | 0.42 ± 0.00 |
| MMLU | acc | 0.23 ±0.00 | 0.23 ±0.00 | 0.24 ±0.00 | 0.23 ±0.00 | 0.23 ±0.00 | 0.25 ±0.00 | 0.23 ±0.00 | 0.23 ±0.00 |
| - humanities | acc | 0.25 ±0.01 | 0.24 ±0.01 | 0.25 ±0.01 | 0.24 ±0.01 | 0.24 ±0.01 | 0.25 ±0.01 | 0.24 ±0.01 | 0.25 ±0.01 |
| - other | acc | 0.24 ±0.01 | 0.24 ±0.01 | 0.24 ±0.01 | 0.24 ±0.01 | 0.24 ±0.01 | 0.27 ±0.01 | 0.25 ±0.01 | 0.24 ±0.01 |
| - social sciences | acc | 0.22 ±0.01 | 0.22 ±0.01 | 0.22 ±0.01 | 0.22 ±0.01 | 0.22 ±0.01 | 0.23 ±0.01 | 0.22 ±0.01 | 0.22 ±0.01 |
| - stem | acc | 0.22 ±0.01 | 0.22 ±0.01 | 0.23 ±0.01 | 0.22 ±0.01 | 0.21 ±0.01 | 0.24 ±0.01 | 0.22 ±0.01 | 0.22 ±0.01 |

| Task | Metric | GPT2-LG | Baseline | Unlearn Baseline | F-MarI | KL-GA | GA | GD | DPO |
|------|--------|---------|----------|------------------|--------|-------|-----|-----|-----|
| WikiText | bits/byte | 0.841 | 0.925 | 0.905 | 0.917 | 0.929 | 26.895 | 0.961 | 0.939 |
|  | byte-pplx | 1.792 | 1.898 | 1.873 | 1.888 | 1.904 | 1.248e+08 | 1.946 | 1.917 |
|  | word-pplx | 22.612 | 30.797 | 28.662 | 29.886 | 31.309 | 1.972e+43 | 35.214 | 32.428 |

Table 5: Comprehensive evaluation results across multiple benchmarks for the GPT2-LG baselines. WikiText is based on perplexity, so lower is better. The higher score is better for all other tests: ARC, PIQA, HellaSwag, and MMLU.

| Task | Metric | LLaMA-3.2-1B | Full Finetune | Unlearn Baseline | F-MarI | KL-GA | GA | GD | DPO |
|------|--------|--------------|---------------|------------------|--------|-------|-----|-----|-----|
| ARC-Easy | acc | 0.65 ± 0.01 | 0.59 ± 0.01 | 0.60 ± 0.01 | 0.58 ± 0.01 | 0.55 ± 0.01 | 0.42 ± 0.01 | 0.57 ± 0.01 | 0.59 ± 0.01 |
|  | acc_norm | 0.61 ± 0.01 | 0.56 ± 0.01 | 0.55 ± 0.01 | 0.56 ± 0.01 | 0.53 ± 0.01 | 0.39 ± 0.01 | 0.55 ± 0.01 | 0.56 ± 0.01 |
| ARC-Challenge | acc | 0.31 ± 0.01 | 0.30 ± 0.01 | 0.31 ± 0.01 | 0.29 ± 0.01 | 0.30 ± 0.01 | 0.27 ± 0.01 | 0.30 ± 0.01 | 0.30 ± 0.01 |
|  | acc_norm | 0.36 ± 0.01 | 0.35 ± 0.01 | 0.36 ± 0.01 | 0.34 ± 0.01 | 0.33 ± 0.01 | 0.29 ± 0.01 | 0.32 ± 0.01 | 0.32 ± 0.01 |
| PIQA | acc | 0.74 ± 0.01 | 0.71 ± 0.01 | 0.71 ± 0.01 | 0.71 ± 0.01 | 0.70 ± 0.01 | 0.63 ± 0.01 | 0.69 ± 0.01 | 0.71 ± 0.01 |
|  | acc_norm | 0.74 ± 0.01 | 0.71 ± 0.01 | 0.71 ± 0.01 | 0.69 ± 0.01 | 0.69 ± 0.01 | 0.62 ± 0.01 | 0.70 ± 0.01 | 0.72 ± 0.01 |
| Hellaswag | acc | 0.48 ± 0.00 | 0.48 ± 0.00 | 0.50 ± 0.00 | 0.47 ± 0.00 | 0.46 ± 0.00 | 0.30 ± 0.00 | 0.45 ± 0.00 | 0.47 ± 0.00 |
|  | acc_norm | 0.64 ± 0.00 | 0.63 ± 0.00 | 0.65 ± 0.00 | 0.62 ± 0.00 | 0.60 ± 0.00 | 0.38 ± 0.00 | 0.59 ± 0.00 | 0.61 ± 0.00 |
| MMLU | acc | 0.37 ± 0.00 | 0.30 ± 0.00 | 0.31 ± 0.00 | 0.28 ± 0.00 | 0.26 ± 0.00 | 0.23 ± 0.00 | 0.26 ± 0.00 | 0.28 ± 0.00 |
| - humanities | acc | 0.35 ± 0.01 | 0.30 ± 0.01 | 0.31 ± 0.01 | 0.29 ± 0.01 | 0.26 ± 0.01 | 0.24 ± 0.01 | 0.27 ± 0.01 | 0.29 ± 0.01 |
| - other | acc | 0.41 ± 0.01 | 0.31 ± 0.01 | 0.32 ± 0.01 | 0.29 ± 0.01 | 0.28 ± 0.01 | 0.24 ± 0.01 | 0.25 ± 0.01 | 0.30 ± 0.01 |
| - social sciences | acc | 0.39 ± 0.01 | 0.32 ± 0.01 | 0.32 ± 0.01 | 0.28 ± 0.01 | 0.26 ± 0.01 | 0.22 ± 0.01 | 0.24 ± 0.01 | 0.26 ± 0.01 |
| - stem | acc | 0.33 ± 0.01 | 0.26 ± 0.01 | 0.28 ± 0.01 | 0.28 ± 0.01 | 0.25 ± 0.01 | 0.21 ± 0.01 | 0.27 ± 0.01 | 0.26 ± 0.01 |

Table 6: Comprehensive evaluation results across multiple benchmarks for the LLaMA-3.2-1B unlearning experiment.

# E   APPENDIX OF SECTION 4.4: DETECTION TESTS

## E.1   DETECTOR METHODS

Here, we provide a more detailed introduction to the current study of copyright content detectors for LLMs so that readers better understand the numerical study in section 4.4. The current study of copyrighted text detectors can be roughly separated into two lines of work:

- **White-box methods:** Perplexity outlier and reference model perplexity outlier [3], domain normalized minimum k-percentage [45], and data-set level inference [29].
  The above methods largely share the same idea of constructing a statistic (or a vector of statistics) that indicates the probability that a model has seen a given sentence or not. It bases the probability on how confidently the model predicts the true output. The idea is based on the intuition that a model that has seen the sentence during training will have high confidence when trying to complete it.
- **Black-box methods:** Direct regurgitation probes [23], Name-cloze membership inference [4], DE-COP: multi-choice preference [8], Output-consistency measures [7], and VeilProbe [18].
  Black-box methods, which do not have access to the model parameters and therefore the output logits or prediction distributions, often use either edit distance (a.k.a. Levenshtein distance) or some token embedding model (e.g. a small transformer) to quantify the distance or similarity between a model's output and a reference string, then generate statistics of the similarity between the two.

Black-box methods are weaker detectors than white-box methods since they do not have access to a model's internals. Since our method assumes access to the model parameter, we tested our method against the current SotA white-box method, the minimum k-percent method [35], to demonstrate the effectiveness of our unlearning in real-world applications.

## E.2 MULTIPLE DETECTION TEST RESULTS

Here, we provide more ablation details on the undetectability result provided in Figure 5, Section 4.4.

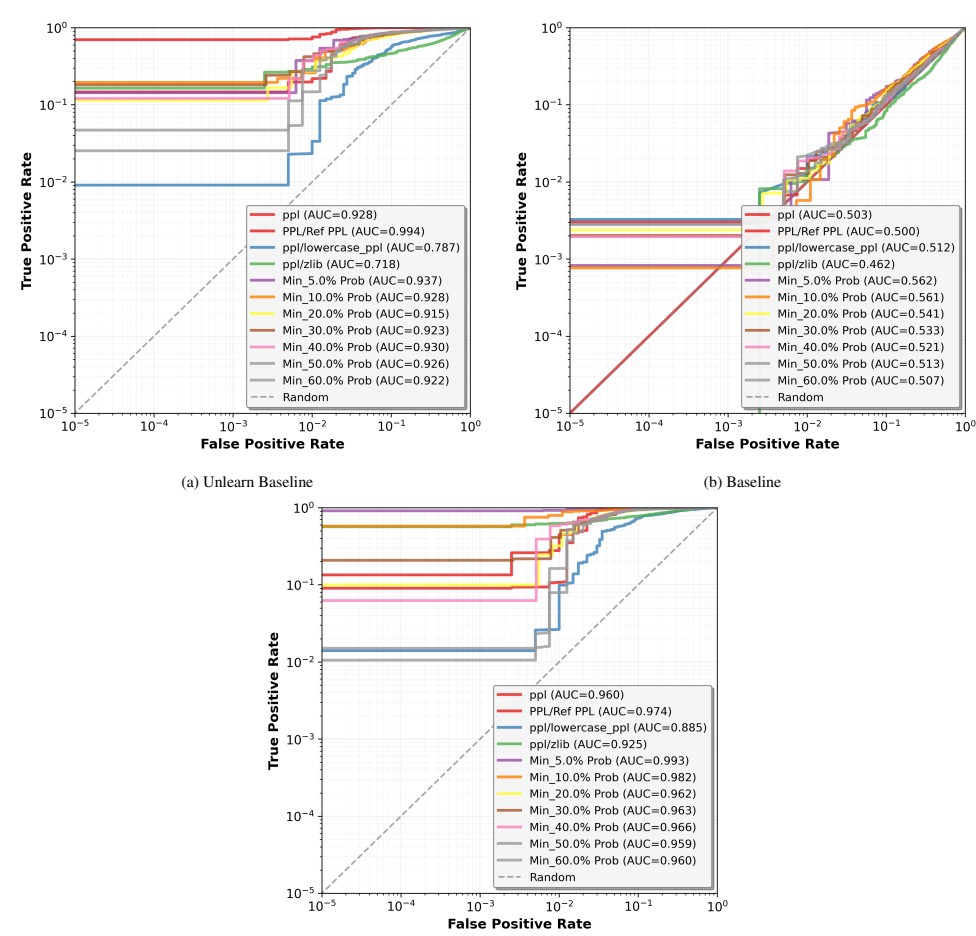

Figure 14: Training data membership detection test of Forgetting-MarI against state-of-the-art detection methods using the 10/90 split Unlearning of the GPT2-LG.

