# OpenReview forum: "Forgetting-MarI: LLM Unlearning via Marginal Information Regularization"
_ICLR.cc/2026/Conference — ICLR 2026 Conference Desk Rejected Submission_

### Official Review · Reviewer_rJHg · 2025-10-26

**Soundness:** 2
**Presentation:** 2
**Contribution:** 2
**Rating:** 2
**Confidence:** 3

**Summary:**

This paper proposes Forgetting-MarI, an LLM unlearning framework that removes only the marginal information contributed uniquely by the data to be forgotten, rather than erasing all related knowledge. It formalizes marginal information via mutual information between retain-only and retain+forget distributions, and introduces a regularization term to penalize this marginal effect during fine-tuning. As a result, the model forgets unwanted data while preserving shared knowledge supported by the retain set. Experiments on GPT-2 and Llama show more stable training, better retain–forget trade-offs, and lower detectability by perplexity or membership inference detectors compared to existing methods like gradient ascent or DPO.

**Strengths:**

1. The paper introduces a principled definition of marginal information using mutual information between retain-only and retain+unlearn distributions, giving a more grounded objective than heuristic gradient-ascent–based approaches.

2. Instead of over-forgetting (removing all related knowledge), it focuses on erasing only the incremental contribution of the unlearned data—better aligned with legal and practical expectations of data removal.

3. Theoretical results bound the residual mutual information and show that after unlearning, confidence-based or perplexity-based detectors cannot reliably detect whether the unlearned data was seen.

**Weaknesses:**

1. The proposed method requires full forward passes over both retain and unlearn sets to estimate token-level or pooled distributions and mutual information, which may be significantly more expensive than lightweight methods. There is no concrete runtime, GPU memory, or scalability analysis for larger models.

2. Although elegant theoretically, estimating JSD across token distributions of two datasets at every update step could be costly and unstable in heterogeneous or long-sequence settings. No analysis is provided for approximation errors or computational shortcuts.

3. Only GPT-2 Large and Llama-3.2-1B are evaluated. It remains unclear how the method performs on realistic industrial-scale LLMs (e.g., 7B, 13B, or 70B) or multilingual/general-purpose datasets.

4. The evaluation focuses on literary text (Harry Potter, Careless People) with artificial retain/unlearn splits. Scenarios like personal data removal, bias removal, or knowledge update (dynamic unlearning) are not extensively tested.

5. The method assumes access to both retain and unlearn datasets during unlearning, which is not always realistic in privacy-sensitive settings (e.g., user-request deletion where only the “forget” data remains).

**Questions:**

1. What is the actual computational overhead of estimating marginal information during training? How does runtime and memory compare to simpler baselines (e.g., GD, DPO, LoRA-based unlearning)?

2. In repeated unlearning requests, does mutual information regularization accumulate errors or drift? Do you observe degradation after many rounds of unlearning?

If these concerns are resolved, I’m open to increasing my score.

---

> ### Author Response · Authors · 2025-11-22
>
> [Question 1]
>
> Thank you for raising this point. We included tables summarizing the real-world computational time of our method compared to others in Appendix D1 of our updated submission.
>
> In theory, since Forgetting-MarI, GD, and LoRA all require a forward pass of both the retain and unlearn sets, the computational cost of them are essentially the same. Forgetting-MarI requires computing p^r, p^u, p^d, and I(X_{MarI};Z). p^r is already calculated for utility (CE or KL), p^u is an additional computation, p^d is the average with negligible cost, I(X_{MarI};Z) is the sample estimation via close-form computation and hence also negligible.
>
> In practice, we added a table that includes the time and peak GPU memory during unlearning. You can find that in Appendix D.1. in the updated version. It shows that the computation cost and the peak GPU memory are similar between KL, DPO, GD and F-MarI.
>
> [Question 2]
>
> Thank you for this important question. We did not observe accumulated errors or drift during unlearning after including the KL term to anchor utility. We added results to our updated submission for a new continual unlearning experiment in Sec. 4.3 that addresses your question.
>
> In addition, from a theoretical perspective, provided a large enough retain set (relatively to the unlearn set), it can be shown that the drift is negligible. This would be an interesting theoretical research direction for future work.
>
> [Weakness 1]
>
> Please see the response to Question 1 or Appendix D1 in the updated version.
>
> In regards to your point about requiring a full forward pass of the retain and unlearn sets, our method does not differ from other full-parameter methods on this.
>
> [Weakness 2]
>
> We use empirical estimation of the mutual information based on p^r and p^u, which are also needed by other unlearning methods, such as KL and GD. Therefore, when compared with existing LLM unlearning methods, the additional computational cost is negligible and we don’t expect any difference in stability when compared to other methods.
>
> For computational cost, we now add the computational time and peak memory comparison table in Appendix D.1. in the updated version to demonstrate similar computational cost between us and the existing methods.
>
> Furthermore, we now provide provable pooling error bound between the pooled MarI loss and token-wise MarI loss as Theorem C.1 in the updated version, and a word-level theoretical guarantee for the pooled MarI as Theorem 3.1 in the updated version.
>
> [Weakness 3] Larger models empirical evidence
>
> As we addressed in response to other reviewers, we are a small group who do not have the computational resources to test on models that would be considered “large” by modern standards. We list using larger models in our conclusion as a future direction, and we would be very curious to see how our method translates to much bigger models. Our hope is that, by sharing our work with the ICLR community, labs and research groups with access to more resources can collaborate with us, try our theoretical framework of marginal information regularization, or develop better marginal information methods.
>
> Here, we want to make it clear again that, despite the fact that we lack the computational resources to try our method on larger models, the theoretical evidence we have indicates that our theoretical framework can easily generalize larger models. Moreover, the empirical evidence from our tests shows that our method outperforms existing methods on mid-scale models at similar computational cost and memory usage.
>
> [Weakness 4]
>
> The proposed framework is designed particularly for removing the marginal information of a given forget set, which could include some of the scenarios you listed.
>
> For personal data, it is not clear what type of personal data you are referring to that our method wouldn’t be equipped to handle. If the personal data comes in the format of text, then our experiments provide sufficient empirical evidence to support the fact that it can handle that type of removal. If the personal data comes in other formats, then our proposed method is not designed for that purpose.
>
> Our method is not designed for unlearning concepts, including bias removal, unless that concept is grounded in a particular piece of text. The same for learning new knowledge. That is not the purpose of the proposed method and our experiments are designed only to provide empirical evidence for the unlearning of text.
>
> [Weakness 5]
>
> Yes, we agree that it is not always realistic to assume access to a true retain set. However, one may use other data as the “retain” set in our setting, provided it provides sufficient information to ground the model’s retained capabilities. There is no practical limitation, even though it is called a “retain” set. We now list this as another future research direction in line 531-532 in the updated version. Thank you for raising this insightful question.

---

### Official Review · Reviewer_BufG · 2025-10-27

**Soundness:** 2
**Presentation:** 1
**Contribution:** 2
**Rating:** 4
**Confidence:** 5

**Summary:**

This paper proposes Forgetting-MarI, a new LLM unlearning method based on marginal information regularization. The key idea is to explicitly penalize the mutual information (MI) between model outputs and the “unlearn indicator,” thereby removing only the marginal contribution of the data to be forgotten ($D_u$) while preserving information supported by the retained data ($D_r$). The authors provide a theoretical formulation of marginal information via MI and JSD divergence, a practical implementation compatible with gradient-based fine-tuning, and experiments on GPT-2 Large and Llama-3.2-1B showing better trade-offs between forgetting and utility compared with Gradient Difference, KL-GA, DPO, and GA.

**Strengths:**

1. Conceptual novelty: The paper introduces a clear and elegant information-theoretic framing of LLM unlearning via marginal information, offering a well-motivated bridge between data privacy and utility preservation.
2. Theoretical rigor: Derivations are mathematically sound, and the theorems (especially Theorems 2.1 and 2.2) link mutual information bounds to empirical detectability in a principled way.
3. Strong empirical trends: Forgetting-MarI outperforms or matches prior unlearning methods in both correlated and uncorrelated settings (Figures 2–5). It also exhibits smoother convergence and better continual unlearning stability.

**Weaknesses:**

1. Experimental scale is limited. All experiments are on small to mid-scale models (GPT-2 Large, Llama-1B). It is unclear whether the proposed method scales to realistic 7B–70B LLMs, where unlearning challenges become severe. The tasks (Harry Potter and Careless People) are relatively narrow and do not demonstrate generalization beyond text completion.
2. Weak connection between theory and empirical validation. The theoretical guarantees rely on information-theoretic quantities (MI, JSD) that are hard to estimate accurately in high-dimensional token distributions. The paper does not empirically verify how MI actually evolves during training or correlate it directly with unlearning success.
3. Limited comparison and ablation. Missing quantitative comparison with recent adapter-based or counterfactual unlearning methods. No ablation on the contribution of the MI loss term.
4. Practical concerns. The computation of per-token JSD may be costly for long sequences; the paper does not discuss training overhead. The method relies on manual tuning of λ, without a principled way to choose it.
5. Poor writing. The submission suffers from the poor organization and writing, especifically the experiment section.

**Questions:**

The author are expected to address the concerns raised in the weakness part, or give reasonable explanation about it.

---

> ### Author Response · Authors · 2025-11-22
>
> Dear reviewer,
>
> We thank the reviewer for insightful questions and acknowledging the novelty of our approach.
>
> [Weakness 1]
>
> We have limited computational resources and have done our best within that limitation. We hope to test our method on larger models, and have that listed as a future direction of potential work.
>
> Regarding your point about the narrowness of the chosen datasets, the theory backing our method is based on model perplexity, and so we focused our efforts on experiments that put our theory to the test. There are so many uncertainties when moving beyond sentence completion, such as prompt difference or model hallucination, that could affect the experimental outcome. (Please see our observation in line 428-431 in the updated version as an example.) So we narrowed our focus when validating our method. Our goal is to avoid, as much as possible, other variables that could affect outputs.
>
> We want to emphasize that all benchmarked unlearning methods are tested equally, hence, the current setup already allows for a fair comparison of general capabilities on a common ground.
>
> [Weakness 2]
>
> Our regularization term is exactly the MI (equivalently JSD), and our method uses the closed-form formula to compute the sample estimation of MI for each gradient descent step. Could you please elaborate on why you believe it is hard to estimate? If you are referring to the sample estimation error, then MI can be well-estimated with a central limit theorem-type of error estimation.
>
> MI is the regularization term, so it decreases with gradient descent and one can explicitly monitor the change of (batch sample estimation) MI since it is the regularization term magnitude. We have added Figure 12 in the updated version to provide a direct visualization of the correlation between the proposed marginal information regularization magnitude and unlearning success.
>
> Regarding the connection between unlearning success and MI, it is theoretically guaranteed in Theorem 2.1 and Theorem 2.2, provided the unlearning success is defined by the perplexity gap which ensures undetectability. But, again, we added a figure (Figure 12 in Appendix D.4.) in the updated version to demonstrate the relationship practically. As an alternative, one can first fix an ideal model perplexity gap (unlearning success threshold), then choose an appropriate hyperparameter (e.g. MI regularizer parameter, learning rate, epoch)  to lower the mutual information to satisfy the ideal gap. This discussion of connection is also added in line 315-321 in the updated version.
>
> [Weakness 3]
>
> We only compare our method with full parameter methods because we want to compare apples to apples. To your point about adapter methods in particular, adapter methods are known to not be able to sustainably handle continual unlearning without accumulated burden, which is one of the main goals underlying our study of marginal unlearning.
>
> For ablation, we do include the ablation study for Forgetting-MarI by including the unlearning trajectory of different choice of parameters in Figure 9 and Figure 10 of the original version. We now state it more clearly in the new version and refer readers to the appendix D.3 for the ablation study.
>
> [Weakness 4]
>
> Please see our response to weakness 2 above. In regards to your point about choosing the regularizer parameter, we mention in our conclusion that this is currently an open problem for all unlearning methods. However, our experiments demonstrate that our method is more stable across choices of λ than other methods as demonstrated in Figure 9, 10 and 13.
>
> [Weakness 5]
>
> We have thoroughly revised parts of our submission, in particular the experimental section 4.2 and 4.3 in the updated version, to make things more clear and easy to follow.

---

### Official Review · Reviewer_1K5z · 2025-11-01

**Soundness:** 3
**Presentation:** 3
**Contribution:** 3
**Rating:** 4
**Confidence:** 4

**Summary:**

This paper presents FORGETTING-MARI, a novel and effective framework for machine unlearning, specifically tailored for Large Language Models (LLMs).

**Strengths:**

1. Relevance and Impact: The paper tackles a highly critical and timely challenge in AI—efficient and effective model unlearning for LLMs. Given the size and computational cost of LLMs, incremental unlearning solutions that preserve utility are essential for real-world deployment and regulatory compliance.

2. Theoretical Clarity: The concept of Marginal Information Regularization is theoretically intuitive and well-aligned with the goal of mitigating catastrophic forgetting. By directly targeting the maximization of retain-set knowledge while minimizing forget-set knowledge, the method provides a targeted optimization objective.

3. Empirical Superiority: The experimental results strongly support the claims, showing that FORGETTING-MARI significantly outperforms existing unlearning techniques in balancing the privacy-utility trade-off. This suggests a more robust and practical solution for LLM unlearning.

**Weaknesses:**

1. Utility Trade-off on General Capabilities: The proposed method exhibits the lowest performance on the MMLU general capability benchmark compared to baselines (Table 5). This suggests a potential trade-off where the high unlearning efficacy might be achieved at the cost of a significant reduction in the model's general utility. A successful unlearning strategy should aim to maximize forgetting while minimally impacting core, general knowledge.

2. Limited Cross-Scenario Generalization and Scalability: The current experimental setup limits the confidence in generalization. Specifically, the two forgetting scenarios ("FORGETTING HARRY POTTER" and "FORGETTING CARELESS PEOPLE") are tested exclusively on different base models (GPT-2 Large and Llama-3.2-1B, respectively). It is recommended to perform cross-scenario evaluation (e.g., FORGETTING HARRY POTTER on Llama-3.2-1B and vice versa) to better demonstrate robustness across different model architectures and tasks.

3. Additionally, the models tested are relatively small (up to 1B parameters); results on larger models (e.g., 7B) are necessary to confirm scalability and practical applicability.

4. In the second subplot of Figure 3, some lines extend beyond the frame of the plot. It is recommended to increase the range of the vertical axis.

**Questions:**

see weakness

---

> ### Author Response · Authors · 2025-11-22
>
> Dear reviewer,
>
> We appreciate the reviewer for the thoughtful review and for acknowledging the novelty of our approach. The goal of the paper is to propose a new unlearning principle that directly targets marginal information. To that end, we provide a rigorous mathematical framework, justification, and guarantee. While we do not have the computational resources to test the proposed method on datasets and models that would be considered “large” by modern standards, we do rigorously test it within our resource capability and observe empirical evidence on mid-scale models that match our claims. Moreover, based on theoretically grounded assumptions, we have no reason to suspect that the theory/method does not generalize to larger-scale models. Therefore, we hope to share the method with the community so that, together as a whole, we can solve the LLM unlearning problem.
>
> [Weakness 1]
>
> We thank the reviewer for raising this point. We expanded our comparison of unlearning methods to include all benchmarks to show the advantage of the proposed method. See Figure 3, Table 5 and Table 6 in the updated version.
>
> Furthermore, we note that high scores on general benchmarks (such as mmlu) do not guarantee meaningful utility: for instance, as shown in Figure 3, GA can outperform all other methods on MMLU despite lacking any utility preservation (exhibiting >2e+43 WikiText perplexity, where exploding perplexity is consistent with theory). Thus, evaluating perplexity or token-level confidence is essential to avoid confounding factors that could affect experiment results.
>
> We included this discussion in the new section General capability (line 418-431) in the updated version.
>
> [Weakness 2]
>
> The choice of which dataset we used for which model was arbitrary. We do not have any reason to believe that our method would perform differently if we performed the cross-validation you’re suggesting, but this could be an avenue for followup work.
>
> Further, we want to emphasize that all benchmarked unlearning methods are tested equally, hence, the current setup already allows for a fair comparison of general capabilities on a common ground.
>
> [Weakness 3]
>
> Reiterating our point above, our experiments were performed within the computational limits of the compute resources we have access to.  We believe the strong alignment between our theory and the empirical results across diverse tests provides robust evidence for the viability and generalizability of our method. But further experimentation on larger models would definitely be a bonus for future work. And we thank the reviewer for highlighting it.
>
> [Weakness 4]
>
> Thank you for pointing this out. We have fixed this in the updated version.

---

### Official Review · Reviewer_uBXm · 2025-11-01

**Soundness:** 1
**Presentation:** 1
**Contribution:** 2
**Rating:** 0
**Confidence:** 3

**Summary:**

The paper proposes Forgetting-MarI, a novel LLM unlearning method that aims to remove the influence of data to be forgotten while preserving knowledge from the retain data. Unlike prior approaches that often over-unlearn information and harm model performance, Forgetting-MarI explicitly penalizes the marginal information contributed by the forget set. This enables the method to provide a provable upper bound on the residual influence of the forgotten data, ensuring undetectability and privacy compliance. Experimental results show that Forgetting-MarI achieves more reliable forgetting and better retention of general model utility compared to existing unlearning methods.

**Strengths:**

- [S1] **Interesting conceptual direction.** The idea of designing an unlearning loss derived from information-theoretic principles is conceptually interesting. The focus on isolating and penalizing marginal information, if made rigorous and well-justified, could offer a new angle for formalizing unlearning objectives in LLMs.

**Weaknesses:**

- [W1] **Poor presentation and clarity.** The paper is difficult to follow overall and would benefit from a substantial reorganization and clearer exposition. Several key issues include:
  - The classification of unlearning methods into “marginal information unlearning” and “full unlearning” is not well-motivated. As the paper itself notes, most existing methods already mix forgetting and retention signals, making them marginal in nature. Therefore, the distinction does not meaningfully clarify what is unique about Forgetting-MarI.
  - Notation and definitions in Section 2 are insufficiently explained. For example, what does $V^T$ represent? How is addition between distributions $p_t^r$ and $p_t^u$ defined? What does $M$ denote in Theorem 2.2? Without clear definitions, the theoretical framework is hard to interpret.
  - The theoretical results in Section 2.2 are difficult to understand and lack intuitive explanations. The implications of the theorems are not well connected to the practical advantages of Forgetting-MarI over other unlearning methods such as gradient ascent or preference-based unlearning.
  - Algorithm 1 is presented in an unreadable form, with long mathematical expressions rather than structured pseudocode. Rewriting the algorithm with modular pseudocode steps would make the method far clearer.
- [W2] **Insufficient and unconvincing experimental design.** The experiments do not convincingly support the claims made in the paper:
  - The Continual Unlearning setup does not actually capture a continual learning scenario. The flat baseline curves in Figure 3 suggest that the experiments were simply run across multiple epochs rather than sequentially unlearning new forget sets, which fails to reflect the continual unlearning problem.
  - The synthetic setup where forget and retain sentences are artificially interleaved is unrealistic and not well motivated. This design weakens the claimed contribution, as real-world unlearning often involves overlapping but distinct datasets (e.g., similar topics or styles), not artificially mixed sequences. The experimental focus on book datasets such as Harry Potter also does not align with the intended motivation of removing marginal information from overlapping textual domains. Benchmarks such as TOFU [A], MUSE [B], or WMDP [C] could be more appropriate and realistic testbeds.
  - For several experiments, F-MarI is compared only against baselines and not existing unlearning methods. A list of such experiments are: general capability measurements with WikiText and MMLU (Tables 4 and 5), another set of general capability measurements on ARC-Easy and PIQA (Table 6), and forget set detection (Figure 6). Without such comparisons, it is difficult to assess the claimed superiority of the proposed method.

[A] TOFU: A Task of Fictitious Unlearning for LLMs. COLM 2024.\
[B] MUSE: Machine Unlearning Six-Way Evaluation for Language Models. ICLR 2025.\
[C] The WMDP Benchmark: Measuring and Reducing Malicious Use with Unlearning. ICML 2024.

**Questions:**

- [Q1] In Section 4.3, the fine-tuning setting cites the TOFU benchmark. Does this mean the experiments followed the TOFU fine-tuning pipeline but used a custom dataset derived from *Careless People: A Cautionary Tale of Power, Greed, and Lost Idealism*?

---

> ### Author Response · Authors · 2025-11-22
>
> Dear Reviewer,
>
> [S1] We would like to first thank the reviewer for confirming that our novel marginal information unlearning provides a principled approach to solve LLM unlearning.
>
> Your review stated, “ if made rigorous and well-justified, could offer a new angle for formalizing unlearning objectives in LLMs.”
>
> In fact, the paper has formalized marginal information, which is a rigorously defined and well-justified unlearning objective. In particular, in our original version, Definition 2.1 provides a mathematically rigorous definition of the marginal information and Proposition 2.1 provides a rigorous justification by using a sharp inequality (meaning the bound cannot be improved without assuming distribution). In addition, Remark 2.1 and Appendix B.2 explain and justify the reason why we chose mutual information and its advantage over other options. Then, Theorem 2.1 and Theorem 2.2 provide mathematically rigorous theoretical guarantees for the unlearning performance against any perplexity, log loss, or model confidence-based detector. All proofs are included in the appendix.
>
> We take the mathematical rigor of our work very seriously. Therefore, we would appreciate clarification on which specific aspects you believe are not rigorous or well-justified, as our contributions clearly demonstrate a solid and coherent framework.
>
> [W1: 1]
>
> We thank the reviewer for raising this important distinction. The original statement of the distinction is stated in the original version: definition 1.1. It states that “Marginal information is the marginal effect on model inference when adding the unlearn set to the retain set.” “Full information unlearning” means eradicating all knowledge contained in the unlearn set.
>
> More specifically, as explained in Appendix A.1 in our original submission, existing LLM unlearning methods are full-information unlearning in nature: their ascent term targets the entire signal of $\mathcal D_u$ (e.g., maximizing CE on $\mathcal D_u$) and they attempt to \emph{indirectly} spare shared/legitimate knowledge by counterbalancing with a retain loss (CE or KL), preference shaping, parameter subtraction, or orthogonality (see details Appendix A.1). We want to emphasize that, there is an intrinsic conflict between the utility and unlearning objective, which is necessary for the counterbalance to work but leads to unstable unlearning and requires extreme effort in parameter-tuning.
>
> In contrast, the proposed methods with “marginal information unlearning” optimize an unlearning objective that \emph{directly} measures and suppresses only the \emph{additional} information contributed by the unlearn set beyond what is already supported by the retain set, and the utility term aims only to stabilize retain performance or help the model learn new datasets. There is no intrinsic conflict between the unlearn and utility objectives.
>
> The above explanation is now added in line 75-86 of our updated submission to further clarify the distinction between methods which are marginal information unlearning in nature or full information unlearning in nature.
>
> [W1: 2]
>
> In line 174 of the original version, we defined V is the vocabulary space, T is the length, so  V^T in line is the standard mathematical notion of the T dimensional Cartesian product space of V, which contains all the possible sequences of vocabulary/token (over V) with length T. p^r_t and p^u_t are defined in lines 176 and 177 which represent the probability density of p^r and p^u at time t. More specifically, p^r_t(v) and p^u_t(v) are the probability of the model inferring v at time t based on r_{<t} and u_{<t} respectively. Therefore, the addition or average of p^r_t and p^u_t  is taken pointwise on v, which is standard notation of probability density function addition and averaging.
>
> M is a constant uniform upper bound of the density ratio we introduced in line 259, and we added an explicit definition of M on line 948 in the updated version of our submission.
>
> [W1: 3]
>
> The implementation/explanation of Theorem 2.2 is stated in lines 255-256 in the original version, which is a generalization of Theorem 2.1 to a neighborhood of u rather than u itself for a stronger nondetectibility guarantee. From our original submission: “Finally, we show that MarI directly controls the score gap even for a neighborhood of u rather than only u itself. Proof can be found in Appendix B.4.”
>
> The practical advantage is that, via Theorem 2.1 and Theorem 2.2, any practitioner can provide provable and auditable unlearning guarantees in terms of non-detectability by simply monitoring the magnitude of the mutual information regularization term during training. This is now included as line 315-321 in the updated version. In comparison, the other methods have no provable guarantee, not to mention monitorable/auditable ones.

---

> ### Author Response · Authors · 2025-11-22
>
> [W1: 4]
>
> We thank the reviewer for raising this important point. We realize that the reduced size of our algorithm in the original version may have made it difficult to read. We want to state that our algorithm in the original version is mathematically clear, rigorous, and well-defined. To address this, we have included a flow chart version in Figure 2 of our updated submission to enhance readability. Additionally, we have provided an enlarged version of the original algorithm in the appendix line 1297-1333 in the updated version to ensure all details are accessible and clear.
>
> [W2: 1]
>
> We thank the reviewer for flagging this important observation. The flat curves you mention are exactly the point of the experiment. While our method shows flat curves, it is clear that other methods demonstrate over-forgetting on the forget set and over-fitting on the retain set. The ideal behavior is to see stabilized retain, unlearn, and validation curves over epochs.
>
> In addition, our updated submission provides new experiments in Section 4.3 in which we use multiple distinct unlearn sets to mimic sequential unlearning requests and we include performance comparisons. It is clear from Figure 4 and Figure 5 that the proposed marginal unlearning method is more robust in this experimental setup as well.
>
>
> [W2: 2]
>
> Our experimental setup is motivated by the case mentioned at the beginning of our paper where two newspapers are reporting the same factual news (line 50-55), and they only differ in minor details. Our choice of alternating sentences is trying to mimic that scenario. It is also well-motivated in practice: you may have the copyright to use one book in the Harry Porter series but not the other. Alternatively, you may have the rights to one publisher’s textbook on a topic but not another’s. The textbooks would cover almost the same technical details, especially in well-established fields, such as mathematics and physics. Our experiments were set up to mimic that scenario. Given that, could you please be more specific about what you mean by “overlapping textual domains”, and why our alternating experiment and correlated experiment are not sufficient for them? Furthermore, for distinct forget and retain sets, in the forgetting Careless People experiment, we purposefully use correlated and uncorrelated sets to further demonstrate the effectiveness of Forgetting-MarI in both overlapping and non-overlapping scenarios.
>
> The goal of the experiment is to test and verify the proposed marginal unlearning framework and ensure that it behaves the same as theory suggests. We believe our experiments provide sufficient empirical evidence to support our claim. Again, if you have a different opinion, please be more specific about why the experiment is not sufficient.
>
> [W2: 3]
>
> Please see Section 4.2 (Figure 3) for a full comparison in the updated version of our submission.
>
> [Q1:]
>
> Our experimental setup follows the general experimental process outlined by TOFU and other unlearn benchmarks. The choice of Harry Porter is due to its widely used unlearn benchmarks, and the choice of Careless People is because it is publicly available and published after the Llama 3.2 model so unlikely to be included in the training set of Llama 3.2.

---

### Note · Program_Chairs · 2026-01-17
**Submission Desk Rejected by Program Chairs**

The following references in this submission do not refer to real documents and/or have major errors in bibliographic information:

 Xiaomin Zhang, Fali Wang, Wenpeng Yin, and Suhang Wang. Domain-calibrated pretraining data detection for large language models. arXiv preprint arXiv:2405.09876, 2024. URL https://arxiv.org/abs/2405.09876.
Jiajun Dong, Wenjuan Han, Tongshuang Wu, and Xiang Ren. Consistency speaks volumes: A black-box test for memorization in LLMs. arXiv preprint arXiv:2403.08999, 2024. URL https://arxiv.org/abs/2403.08999.
Zihan Hu, Yuo Xu, Lavanya Shukla, and Milad Nasr. VeilProbe: Automated black-box membership detection for large language models. arXiv preprint arXiv:2406.01840, 2024. URL https://arxiv.org/abs/2406.01840.
Rahul Maini, Nicholas Carlini, Kobbi Nissim, and Shafi Goldwasser. When one example fails, a thousand may succeed: Dataset-level membership inference for large language models. In International Conference on Machine Learning (ICML), 2024. URL https://arxiv. org/abs/2403.0123
Maria Karamolegkou, Emily Dinan, and Angela Fan. Leaks in the library: Prompting large language models to reveal copyrighted text. In Findings of EMNLP, 2023. URL https: [//aclanthology.org/2023.findings-emnlp]//aclanthology.org/2023.findings-emnlp. 456.
João Duarte, Diego Esteves, and André F. T. Martins. DE-COP: Detecting copyrighted passages in large language models via multi-choice paraphrase tests. In Proceedings of the 62nd Annual Meeting of the Association for Computational Linguistics (ACL), 2024. URL https://arxiv.org/abs/2404.07111.